# The Polar Front in the northwestern Barents Sea: structure, variability and mixing

Eivind H. Kolås[1], Ilker Fer[1], and Till M. Baumann[1]

[1]Geophysical Institute, University of Bergen and Bjerknes Center for Climate Research, Bergen, Norway

**Correspondence:** Eivind H. Kolås (eivind.kolaas@aqua-kompetanse.no)

**Abstract.**

In the northwestern Barents Sea the warm and salty Atlantic Water meets the cold and fresh Polar Water, forming a distinct thermohaline front (the Barents Sea Polar Front). Here we present the structure of the front, its variability and associated mixing using observations from two cruises conducted in October 2020 and February 2021 during the Nansen Legacy project, in the region between Hopen Trench and Olga Basin. Ocean stratification, currents, and turbulence data were obtained during seven ship transects across the Polar Front near 77°N, 30°E. These transects are complemented by four missions using ocean gliders, one of which was equipped with microstructure sensors to measure turbulence. Across the front, we observe warm ($>1°C$) and salty ($>35.0\,\mathrm{g\,kg^{-1}}$) Atlantic Water intruding below the colder ($< 0°C$) and fresher ($<34.6\,\mathrm{g\,kg^{-1}}$) Polar Water, setting up a baroclinic front and geostrophic currents reaching $25\,\mathrm{cm\,s^{-1}}$, with estimated eastward transport of $0.2 \pm 0.6\,\mathrm{Sv}$ ($1\,\mathrm{Sv} = 1 \times 10^6\,\mathrm{m^3\,s^{-1}}$). We observe anomalous warm and cold-water patches on the cold and warm side of the front, respectively, collocated with enhanced turbulence, where dissipation rates of turbulent kinetic energy range between $10^{-8}$ and $10^{-7}\,\mathrm{W\,kg^{-1}}$. Short-term variability below the surface mixed layer arises from tidal currents and mesoscale eddies. While the effects of tidal currents are mainly confined to the bottom boundary layer, eddies induce significant shifts in the position of the front, and alter the isopycnal slopes and the available potential energy of the front. Substantial water mass transformation is observed across the front, likely a result of eddy-driven isopycnal mixing. Despite the seasonal changes in the upper layers of the front (0–100 m) influenced by atmospheric forcing, sea ice formation, and brine rejection, the position of the front beneath 100 m depth remained relatively unperturbed.

## 1  Introduction

The Barents Sea is one of the main gateways through which Atlantic Water (AW) enters the Arctic Ocean (Figure 1). About 2 Sv ($1\,\mathrm{Sv} = 1 \times 10^6\,\mathrm{m^3\,s^{-1}}$) of AW enters through the Barents Sea Opening in the west, continuing as a topographically steered current along the Bear Island Trough (Loeng, 1991; Smedsrud et al., 2010). Upon reaching the Central Bank, about 1 Sv of AW continues north along the Hopen Trench towards the Great Bank (Kolås, 2024, Ch. 4). Polar Water (PW) enters from the north occupying most of the Great Bank and the Spitsbergen Bank (Loeng, 1991; Lien et al., 2017). The location where these two water masses meet is named the Barents Sea Polar Front (Loeng, 1991; Harris et al., 1998; Oziel et al., 2016), and is a

site for water mass transformation with significance both for the overturning circulation and ventilation of the Arctic Ocean (Årthun et al., 2011). Water masses are defined in Section 3.1, Table 3.

The Barents Sea Polar Front, which we refer to as the Polar Front (PF), is part of the North Polar Frontal Zone, a major climatic feature in the Nordic Seas where the characteristic scale of temperature and salinity variability is 5–50 km (Rodionov, 1992). The frontal zones are highly dynamical regions with mesoscale features ranging from a few to hundreds of kilometers,

including, but not limited to; fine structures due to local wind, precipitation, internal waves, isolated eddies, frontal meanders and advective intrusions (Rodionov, 1992; Gula et al., 2022).

The PF is mainly topographically steered, particularly in the western Barents Sea from Bear Island along the Spitsbergen Bank, to the Great Bank (for place names see Fig. 1), where it follows the 200-250 m isobath (Johannessen and Foster, 1978; Gawarkiewicz and Plueddemann, 1995; Oziel et al., 2016; Kolås et al., 2023). However, the climate of the Barents Sea has

been shown to affect the position of the front south of Bear Island; in warmer periods with stronger winds, the front shifted upslope compared to colder periods (Ingvaldsen, 2005). Around the Central Bank, and in the eastern Barents Sea, the PF is not as stationary as along the Spitsbergen Bank, covering a broader zone of mixing (Oziel et al., 2016).

There have been few detailed studies of the frontal structure in the Barents Sea. The Barents Sea Polar Front Experiment studied the front at the southern flank of the Spitsbergen Bank in summer 1992 (Gawarkiewicz and Plueddemann, 1995;

Parsons et al., 1996). The upper 20–40 m of the PF is characterized by a clear horizontal salinity gradient separating Arctic meltwater in the north from Atlantic surface water in the south. Because the horizontal temperature gradient is weak in the surface mixed layer, the surface front manifests as a density front, with a maximum density difference across the front of about $0.8\,\mathrm{kg\,m^{-3}}$ (Parsons et al., 1996). Between the surface layer and 100 m depth, the front is defined by a horizontal temperature gradient and contains numerous fine structure intrusions (Parsons et al., 1996). Below 100 m from the surface, the gradients

are weaker, likely due to bottom boundary mixing (Fer and Drinkwater, 2014), resulting in a broader frontal zone. In addition, the horizontal temperature and salinity gradients below 100 m depth compensate for their contribution to density, resulting in a barotropic front with horizontal density lines (Gawarkiewicz and Plueddemann, 1995; Parsons et al., 1996). While other observations such as those by Johannessen and Foster (1978), Fer and Drinkwater (2014) and Våge et al. (2014) agree on the overall structure of the PF, the circulation and interaction of AW and PW across the front remains poorly understood.

Direct measurements of turbulent mixing along the PF suggest that diapycnal mixing is mainly confined to the boundary layers during strong tidal currents and/or strong wind events (Sundfjord et al., 2007; Fer and Drinkwater, 2014). Using detailed measurements of ocean turbulence, hydrography, currents, and nutrients from the PF across Spitsbergen Bank near Hopen in May 2008, Fer and Drinkwater (2014) observed enhanced biological activity between a density compensated thermohaline front near the 150-m isobath and a tidal front on the cold side of the PF. Tidal currents along the slope of the Spitsbergen

Bank can have a peak-to-peak variability of $1\,\mathrm{m\,s^{-1}}$, and bottom boundary dissipation rates reach $10^{-6}\,\mathrm{W\,kg^{-1}}$ (Fer and Drinkwater, 2014). However, on the warm side of the PF, away from the boundary layers, waters are in general quiescent, and mixing mainly occurs along isopycnals with interleaving layers of AW and PW (Fer and Drinkwater, 2014; Våge et al., 2014). Using concomitant measurements of turbulence and biogeochemical variables collected during the same October 2020 cruise reported here, Koenig et al. (2023) estimated a substantial transfer of nitrate and dissolved inorganic carbon across the PF

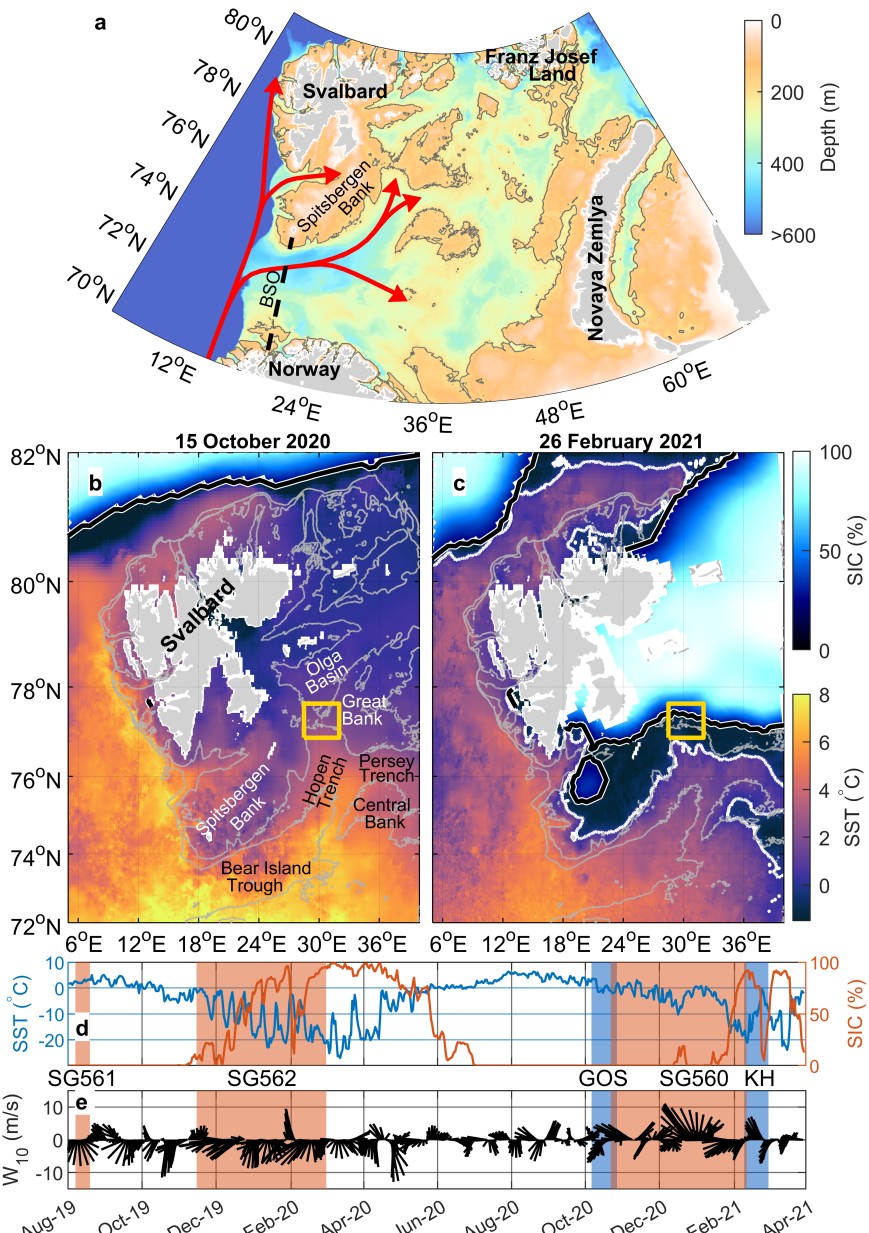

**Figure 1.** (a) Overview map of the Barents Sea. Gray contour indicates the 200 m isobath. Red arrows outline the main Atlantic Water pathways. Black dashed line shows the Barents Sea Opening (BSO). (b) and (c) Zoom in on the northwestern Barents Sea at the time of the fall and winter cruise, respectively, showing sea surface temperature (SST) and sea ice concentration (SIC). Black line indicates the sea ice edge where SIC is 15%. Isobaths are drawn at 200 and 350 m depth. Yellow rectangles show the frontal region studied. White line in (c) indicates the 0°C isotherm in the SST data. SIC is from EUMETSAT OSI-SAF (2017). SST is from the product SEAICE_ARC_SEAICE_L4_NRT_OBSERVATIONS_011_008 from Copernicus (2019). (d) SST and SIC spatially averaged over the yellow box in (b). (e) Daily mean winds at 10 m height (W$_{10}$) spatially averaged over the yellow box in (b), and smoothed over 7 days. Glider and cruise periods are highlighted in red and blue, respectively (SG: Seaglider, GOS; RV G.O. Sars; KH: RV Kronprins Haakon).

from the Atlantic domain to the Arctic domain. Vertical fluxes driven by Ekman pumping were of similar order as the vertical turbulent fluxes.

Another source for mixing along the front can be mesoscale eddies. Eddies detaching from a mean flow diffuse properties along isopycnal surfaces, redistributing heat, salt, and nutrients (Mcwilliams, 2008). Eddies are known to play a key role in distributing AW heat in the Arctic Ocean (von Appen et al., 2022), particularly along the AW inflow west and north of Svalbard

(von Appen et al., 2016; Hattermann et al., 2016; Crews et al., 2018; Våge et al., 2016). In the Barents Sea, satellite radar images from 2007 and 2011 indicated that eddies were frequently generated near the PF southwest of Svalbard (Atadzhanova et al., 2018). However, the role of these eddies in distributing and mixing AW in the Barents Sea is not clear. Porter et al. (2020) observed a cold-core surface eddy south of the PF in the Hopen Trench, and traced its origin back to the north of the PF using sea surface height from satellites. Combining their observations with that of Atadzhanova et al. (2018), they estimated that

the southward transport of cold and fresh water by eddies forming in the PF region was about 0.35 Sv. In addition, Porter et al. (2020) state that this transport is likely an underestimate as eddies in the Barents Sea PF region tend to be masked from satellite-derived sea surface temperature (SST) observations due to thermal capping. The role of eddy-driven along-isopycnal mixing of AW across the PF in the Barents Sea requires further investigation.

The purpose of this paper is to describe the hydrography, dynamics, and mixing in the vicinity of the PF, using observations

from multiple repeat transects over different timescales from tidal cycle to seasons. The dataset coverage is a significant advance relative to earlier studies. The specific location studied is the sill between the Hopen Trench and the Olga Basin (yellow square, Figure 1b). This region is of particular interest as the AW flowing north towards this sill is able to cross below the PF and continue northwards as a topographically steered current (Kolås, 2024, Ch. 4).

## 2 Data

Observational data from two scientific cruises from fall 2020 and winter 2021 (Fer et al., 2023e, b, c, d, a) are supplemented by four glider missions (Kolås et al., 2022) covering the Barents Sea Polar Front in the period from 2019 to 2021. An overview of the data and the corresponding source to access the data are listed in Table 1. All data were collected as part of the Nansen Legacy project. The data presented here is a subset of a larger dataset reported in Kolås et al. (2023, preprint), which addressed the AW circulation and hydrography in the northwestern Barents Sea, and compared it to historical data. An overview of the

data coverage across and near the front is shown in Figure 2. Platform-specific details are given in the following subsections.

### 2.1 Hydrographic Measurements From Cruises

The cruises were conducted onboard the Research Vessel (RV) *G.O. Sars* between 6 and 27 October 2020 (cruise report: Fer et al., 2021) and the icebreaker RV *Kronprins Haakon* between 9 February and 1 March 2021 (cruise report: Nilsen et al., 2021). We will call them the fall cruise and the winter cruise, respectively. Conductivity-temperature-depth (CTD) profiles

during both cruises were collected using a Sea-Bird Scientific, SBE 911plus CTD system, with a 200 kHz Benthos altimeter allowing measurements close to the seabed. The CTD system was equipped with a SBE 32 Carousell fitted with bottles for

**Table 1.** Overview of data used in this study. See text for platform, instrument, and sensor descriptions.

| Platform | Start-End | Instrument or Sensor | Access |
|---|---|---|---|
| sg561 | 7 August 2019 – 19 August 2019 | CTD, DAC | Kolås et al. (2022) |
| sg562 | 15 November 2019 – 1 March 2020 | CTD, DAC | Kolås et al. (2022) |
| *G.O. Sars* | 6 October 2020 – 27 October 2020 | CTD, LADCP, SADCP | Fer et al. (2023a) |
| | | MSS | Fer et al. (2023e) |
| *Odin* | 8 October 2020 – 24 October 2020 | CTD, DAC | Kolås et al. (2022) |
| | | MR | Fer et al. (2023c) |
| sg560 | 22 October 2020 – 11 February 2021 | CTD, DAC | Kolås et al. (2022) |
| *Kronprins Haakon* | 9 February 2021 – 1 March 2021 | CTD, LADCP, SADCP | Fer et al. (2023d) |
| | | MSS | Fer et al. (2023b) |

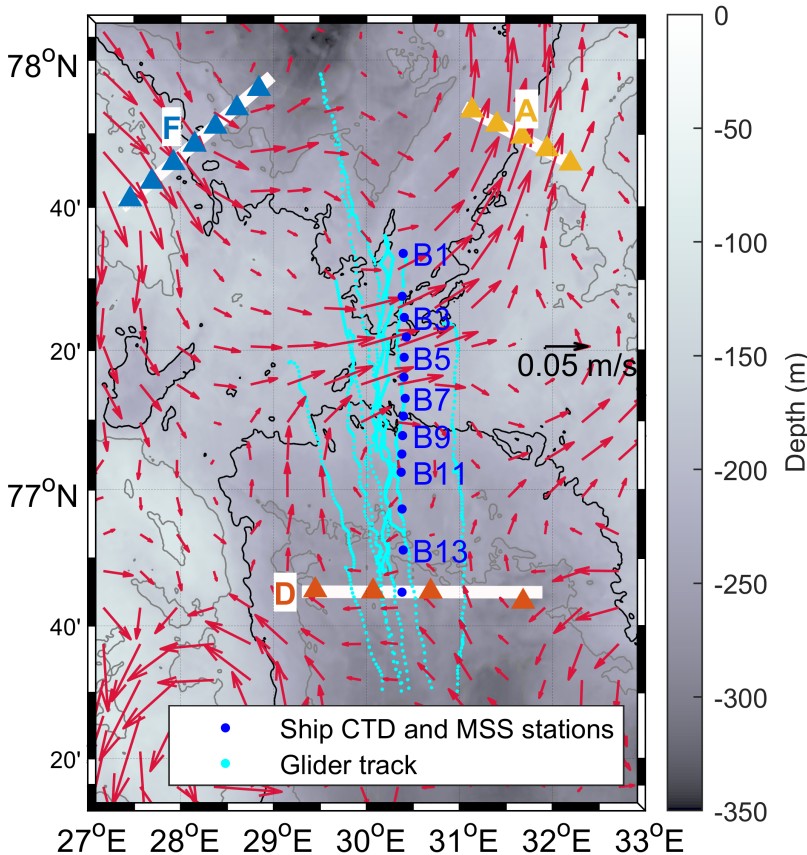

**Figure 2.** Overview of the data coverage across the PF on the sill between Hopen Trench (south) and Olga Basin (north). Isobaths are shown at every 50 m between 50 m and 300 m and the 200 m isobath is highlighted in black. Red quivers are a subset of the objectively mapped, divergence-free, depth-average currents presented in Kolås et al. (2023, preprint). Black quiver shows the scale. White sections (A,D and F) with adjoining triangles show the location of the CTD profiles presented in Figure 9.

collecting water samples at all stations. Bottle samples are analyzed using a Guildline Portasal 8410 salinometer, and used to calibrate salinity. 64 and 89 CTD profiles were collected during the fall and winter cruises, respectively. Pressure, temperature, and practical salinity data are accurate to $\pm 0.5$ dbar, $\pm 2 \times 10^{-3}\,°C$, and $\pm 3 \times 10^{-3}$, respectively. CTD data were processed using the standard SBE Data Processing software. Conservative Temperature, $\Theta$, and Absolute Salinity, $S_A$, were calculated using the thermodynamic equation of seawater (IOC et al., 2010), and the Gibbs SeaWater Oceanographic Toolbox (McDougall and Barker, 2011). Cruise CTD data are accessible from Fer et al. (2023a) and Fer et al. (2023d).

## 2.2 Current Profiles From Cruises

The CTD frames on both vessels were fitted with a pair of acoustic Doppler current profilers (ADCPs), so-called lowered-ADCPs (LADCPs). The LADCPs were 300 kHz Teledyne RD Instruments (RDI) Sentinel Workhorses, one mounted pointing downward and one upward. The LADCPs were synchronized and set to provide data vertically averaged in 8 m bins. On *G.O. Sars* (fall cruise), the LADCPs had internal batteries, while on *Kronprins Haakon* (winter cruise) they had an external battery mounted on the frame. Compasses were calibrated on land prior to cruises with resulting errors less than 1–2° for the *G.O. Sars* cruise (fall), and errors less than 4° for the *Kronprins Haakon* cruise (winter). LADCP data were processed using the LDEO software version IX-13 (Visbeck, 2002). The LADCP profiles were constrained by navigation data and profiles from the ship-mounted ADCPs (SADCP).

Both research vessels had Teledyne RDI Ocean Surveyor SADCPs operating at one or two acoustic frequencies. Here we use the 150 kHz SADCP on *Kronprins Haakon* (winter) and the 75 kHz SADCP on *G.O. Sars* (fall). The SADCP on *Kronprins Haakon* was flush-mounted in the hull and protected by an acoustically transparent window allowing for profiling when moving through ice. The 150 kHz SADCP collected profiles in 4 m bins with 4 m blanking distance, using narrowband mode for optimal range, while the 75 kHz SADCP was set to measure in 8 m bins with 8 m blanking distance. SADCP data were collected using the onboard VmDAS software or the University of Hawaii data acquisition software, depending on the vessel. Post-processing was done using the CODAS software maintained by the University of Hawaii, to an uncertainty of 2–3 cm s$^{-1}$ (Firing and Ranada, 1995). Current profiles from the fall and winter cruises are accessible from Fer et al. (2023a) and Fer et al. (2023d), respectively.

## 2.3 Glider Data

Three Kongsberg Seagliders and one Teledyne Webb G3 Slocum glider conducted transects across the PF in the study region in the Barents Sea in the period from 2019 to 2021. Gliders are buoyancy-driven autonomous underwater vehicles. Both gliders control their pitch by moving an internal battery pack forward and aft along the longitudinal axis of the glider. Details about the individual missions are shown in Table 2. Here we only use the part of the glider data that is within 80 km of the PF location (Figure 2). The complete mission data are reported and analyzed in Kolås et al. (2023, preprint). The typical horizontal distance between two surfacing locations was 1 km. The gliders operated between the surface and seabed, sampling CTD on both ascents and descents. For each dive, a depth-averaged current (DAC) was estimated based on the deviation between the expected surfacing location deduced from the internal flight model and the actual surfacing location. During post-processing,

**Table 2.** Summary of transects collected across the PF. Salinity data from the ship CTD during cruises are corrected against water sample analyses (not listed; see the cruise reports). Offsets in salinity and temperature applied to the different glider missions to correct the glider profiles against the ship CTD are listed. For the glider data, only profiles within 80 km of the Polar Front are included. The duration of ship transects (in hours) is provided in the parenthesis.

| Platform | Transect Start-End | Number of profiles | Offset applied to glider salinity / temperature |
|---|---|---|---|
| sg561 | 13-19 August 2019 | 154 | 0.016 / 0.082°C |
| sg562 | 15-19 November 2019 | 172 | −0.005 / −0.060°C |
|  | 5-11 December 2019 | 172 |  |
|  | 13-19 December 2019 | 188 |  |
| *G.O. Sars* | 14-15 October 2020 (12 h) | 17 |  |
|  | 15 October 2020 (8 h) | 11 |  |
|  | 15 October 2020 (6 h) | 10 |  |
|  | 15 October 2020 (7 h) | 11 |  |
|  | 17-18 October 2020 (8 h) | 15 |  |
| *Odin* | 10-16 October 2020 | 220 | 0.014 / 0.054°C |
|  | 16-21 October 2020 | 278 |  |
|  | 21-24 October 2020 | 192 |  |
| sg560 | 1-11 November 2020 | 290 | 0.017 / −0.209°C |
|  | 11-23 November 2020 | 302 |  |
| *Kronprins Haakon* | 15-16 February 2021 (14 h) | 24 |  |
|  | 16 February 2021 (17 h) | 20 |  |

each temperature and salinity profile was despiked. A data point exceeding twice the root mean square (rms) value of $(x - xs)$, where $x$ is the profile data and $xs$ is a 5-point median filter was removed. In addition, at each pressure level, standard deviation is calculated over all profiles of that mission, and outliers exceeding three standard deviations from a 2D smoothed field of the data were removed. Salinity and temperature offset correction were applied after comparing the deep part of glider dives to nearby CTD profiles collected from research vessels. We expect insignificant instrumental drift in glider measurements over such short sampling durations (several months) and argue that one offset value is sufficient. The offsets applied to the different missions are listed in Table 2.

### 2.3.1 Seagliders

Each Seaglider was equipped with a Kistler piezoresistive pressure sensor, a SBE CT Sail and an Aanderaa dissolved oxygen sensor. The Seagliders operated with a vertical velocity close to $8\,\mathrm{cm\,s^{-1}}$, and sampling rates normally aimed to sample conductivity and temperature every meter while oxygen was sampled every 5 m. The Seaglider data sets were processed using the University of East Anglia Seaglider toolbox, based on the methods described by Garau et al. (2011) and Frajka-Williams

et al. (2011). Manual flagging was applied to the salinity and temperature profiles during processing. Processed $S_A$ and $\Theta$ are accurate to $0.01\,\mathrm{g\,kg^{-1}}$ and $0.001°\mathrm{C}$, respectively, and DAC is accurate to $0.01\,\mathrm{m\,s^{-1}}$ (Seaglider, 2012, p. 9). Despite the thermal-lag correction applied to the records from the unpumped CT sensors of the Seagliders, some profiles were noisy with spurious overturns. During post-processing we removed instabilities where the absolute difference between the density profile and the sorted-density profile was larger than $0.02\,\mathrm{kg\,m^{-3}}$. The number of data points removed was less than $1\,\%$ of the data.

For these missions, Seagliders were operated with the ice-avoidance algorithm. Occasionally, northerly winds pushed sea ice over the gliders, preventing them from surfacing to obtain a GPS fix. These events normally lasted for only a few dives, yet on two occasions the glider remained under sea ice for longer than 24 hours, causing the glider to go into "escape mode", heading south. The longest stretch without a GPS fix lasted for 34 dives or about 44 hours. These dives lacked GPS fixes, hence DAC was not estimated. Longitude and latitude were linearly interpolated between the last known position before encountering ice and the first known position after the glider exited ice. In the two cases where the glider turned south while under the ice (in the escape mode), the northernmost position was extrapolated using the last available good horizontal velocity estimate of the glider, the compass heading, and the depth of the seabed. More details on the processing and glider deployments are given in the data description in Kolås et al. (2022).

### 2.3.2   Slocum gliders

The Slocum glider, *Odin*, carried a pumped SBE CTD sensor (CTD41CP). The glider data were processed using the software developed by G. Krahmann (GEOMAR; Krahmann, 2023). This includes correction for thermal inertia of the conductivity cell and a hydrodynamic model from which the angle of attack and flow rate past the sensor are computed. Final profiles have a horizontal along-track resolution of about 0.5 km and vertical bins of 1 m.

*Odin* additionally carried a turbulence package for measuring small-scale shear across the Polar Front. The turbulence package is described in section 2.4. Care was taken to minimize vibration noise in the vehicle in order not to contaminate the turbulence measurements. The battery was set to fixed mode, preventing the pitch motor from running during the glide. Profiles were kept symmetrical using autoballast control to command pump volumes for diving and climbing, targeting a vertical velocity of $15\,\mathrm{cm\,s^{-1}}$. The glider was configured to inflect 15 m above the seabed. In order to capture the complete water column, particularly the top few meters, the glider carried out each ascent up to the surface before starting a new dive. The depth initiating the surfacing procedure was set to 0 m in order to avoid contamination from the air bladder and ballast pump that automatically turns on during the surfacing procedure.

### 2.4   Turbulence measurements from glider

*Odin* was equipped with a turbulence package, an integrated MicroRider-1000LP (MR) from Rockland Scientific, Canada. The MR was attached on top of the glider, similar to the setup described by Fer et al. (2014). It was powered by the internal battery of the glider, and could thus be remotely turned on and off. Data were stored internally on a compact flash memory card. All turbulence sensors protruded about 25 cm from the nose of the glider, measuring turbulence outside the deformed flow field due

to the glider. Unfortunately, the MR malfunctioned on 15 October due to a bad batch of CR123 battery cell, and consequently, only one transect across the PF was obtained with the MR.

The MR was equipped with two airfoil velocity shear probes (SPM-38), two fast-response thermistors (FP07), a pressure transducer, a two-axis vibration sensor (a pair of piezo-accelerometers), and a high-accuracy dual-axis inclinometer. The MR samples the signal plus signal derivatives on the thermistor and pressure transducer, and the derivative for shear signals, allowing high-resolution measurements. The sampling rate is 512 Hz for the vibration, shear and temperature sensors, and 64 Hz for pitch, roll, and pressure. The accuracy of the measurements is 0.1% for the pressure, 2% for the piezo-accelerometers and 5% for the shear probes.

The MR data was processed and published following the guidelines of the ATOMIX working group (https://wiki.app.uib.no/atomix/index.php/Main_Page, Lueck et al., 2024). The angle of attack and the flow past the sensor used in processing the shear probe data are obtained from the hydrodynamic flight model of the glider. The method for dissipation estimates is outlined in section 3.3. Data are accessible from Fer et al. (2023c).

## 2.5 Turbulence measurements from ship

During both cruises, ocean microstructure measurements were made using the long version of the Microstructure Sensor Profiler, MSS90L from Sea&Sun Technology, Germany. The MSS is a loosely-tethered free-fall instrument equipped with two airfoil probes, a fast-tip thermistor (FP07), a vibration sensor and conventional CTD sensors for precision measurements. The sensors point downward when the instrument profiles vertically, and all sample at a rate of 1024 Hz. The instrument is ballasted for a typical fall speed of $0.6 - 0.7 \, \mathrm{m \, s^{-1}}$ and is decoupled from operation-induced tension by paying out cable at sufficient speed to keep it slack. Data are transmitted in real-time to a ship-board data acquisition system. In total, 207 and 172 profiles were collected during the fall and winter cruise, respectively. Casts were made down to about 5–15 m height above the bottom. Occasionally the profiler landed at the bottom. A sensor protection guard at the leading end of the profiler prevented damage to the sensors.

The MSS data were processed and published following the guidelines of the ATOMIX working group (Lueck et al., 2024). The method is outlined in section 3.3. MSS data from the fall and winter cruises are accessible from Fer et al. (2023e) and Fer et al. (2023b), respectively.

## 2.6 Other data

Hourly data of wind at 10 m and air temperature at 2 m are from the ERA5 reanalyses (Hersbach et al., 2018). Wind stress is calculated as $(\tau_x, \tau_y) = \rho_{\mathrm{air}} C_D |\mathbf{u}_{10 \, \mathrm{m}}| (u_{10 \, \mathrm{m}}, v_{10 \, \mathrm{m}})$, where $\rho_{\mathrm{air}}$ is the density of air, $u_{10 \, \mathrm{m}}$ and $v_{10 \, \mathrm{m}}$ are the eastward and northward components of wind velocity at 10 m, and $C_D$ is the drag coefficient calculated as a function of sea ice concentration (SIC) following equation 22 in Lüpkes and Birnbaum (2005). Time series of wind velocity, wind stress and air temperature are averaged over the region 28.5°E to 32.0°E, 76.90°N to 77.65°N (yellow box in Figure 1b).

Sea ice concentration on 15 October 2020 and 26 February 2021 is from EUMETSAT OSI-SAF (2017). Sea surface temperature is from the product SEAICE_ARC_SEAICE_L4_NRT_OBSERVATIONS_011_008 at 0.05° resolution based upon

**Table 3.** Water mass definitions following Sundfjord et al. (2020), using Conservative Temperature, $\Theta$, Absolute Salinity, $S_A$, and potential density anomaly, $\sigma_0$. Note that our definition of the warm Polar Water, includes only waters with $\sigma_0 \geq 27.8$ in order to exclude surface waters from the wPW.

| Water mass | $\Theta$ (°C) | $S_A$ (g kg$^{-1}$) | $\sigma_0$ (kg m$^{-3}$) |
|---|---|---|---|
| Atlantic Water (AW) | $\geq 2$ | $\geq 35.06$ | |
| Polar Water (PW) | $< 0$ | | $< 27.97$ |
| warm Polar Water (wPW) | $\geq 0$ | $< 35.06$ | $\geq 27.8$ |
| modified Atlantic Water (mAW) | $0 \leq \Theta < 2$ | $\geq 35.06$ | |

observations from the Metop_A AVHRR instrument (Copernicus, 2019). Bathymetry data are from the International Bathymetric Chart of the Arctic Ocean Version 4.0 (IBCAO-v4) (Jakobsson et al., 2020).

Daily and monthly sea level anomalies (SLA) and surface geostrophic velocity anomalies derived from the SLA are from
205 the product SEALEVEL_GLO_PHY_L4_MY_008_047 at 0.25° resolution (Copernicus, 2023). This is a reprocessed product using the DUACS multimission altimeter data processing system (Pujol et al., 2016; Carrere et al., 2016). Different altimeter missions are merged and optimally interpolated to obtain SLA with respect to a twenty-year (1993–2012) mean.

Tidal currents are obtained from the Arctic 2 km Tide Model (Arc2kmTM). This is a barotropic ocean tide model on a 2x2 km grid that provides tidal ellipse properties in the Arctic Ocean domain for 4 semidiurnal (M2, S2, K2, N2) and 4 diurnal
(K1, O1, P1, Q1) tidal constituents (Howard and Padman, 2021).

## 3 Methods

### 3.1 Water masses

The water masses used in this study are listed in Table 3, and follow Sundfjord et al. (2020). These definitions are based on previous water mass definitions in literature such as Lind et al. (2018); Loeng (1991); Rudels et al. (2005). However, we have
215 made a modification to the definition of warm Polar Water (wPW), by including only waters with potential density anomaly $\sigma_0 \geq 27.8 \, \text{kg m}^{-3}$. By excluding wPW with $\sigma_0 < 27.8 \, \text{kg m}^{-3}$ we only consider the wPW which is a mixture between AW and PW, excluding surface waters influenced to a greater extent by seasonal processes such as atmospheric heating and ice melting. The reason for this is that we miss the surface front (upper 50 m) in all ship transects, and capture it only in a few of the glider transects. The upper 50 m and the water column below are two different domains that are influenced by different
physical processes. Here we mainly consider the lower layer.

### 3.2 Hydrographic sections and geostrophic flow

In order to remove fine-scale variability and obtain fields representative of the background hydrography and geostrophic currents, we produce objectively mapped sections. Along the ship transects, we define a 1 km by 1 m (horizontal by vertical)

resolution grid. We objectively interpolate the hydrography data from the ship's CTD system and precision temperature and conductivity sensors of the MSS, and the $u$ and $v$ components of the ocean currents measured by the SADCP, onto the grid using a two-dimensional Gaussian covariance function: $\mathrm{cov}(x,z) = e\delta(x,z) + (1-e)\exp(-x^2/L_x^2 - z^2/L_z^2)$ with $\delta$ being the Dirac function, $e = 0.05$ the relative error, and correlation scales $L_x = 30\,\mathrm{km}$ and $L_z = 30\,\mathrm{m}$. Correlation length scales are estimated based on a semi-variogram analysis, similar to that described in appendix B in Kolås et al. (2020).

Gliders are affected by strong currents that hinder them from collecting data along a straight transect. For the glider data, we first define a transect line by using the best linear fit to the glider surfacing locations. Next, we objectively map the glider $\Theta$ and $S_A$ at each depth level (1 m vertical resolution), and the glider DAC, horizontally onto the new transect at 1 km horizontal resolution, using $\mathrm{cov}(x,y)$, and $L_x = 30\,\mathrm{km}$ and $L_y = 30\,\mathrm{km}$. Finally, we objectively map $\Theta$ and $S_A$ along the transect using $\mathrm{cov}(x,z)$, and $L_x = 30\,\mathrm{km}$ and $L_z = 30\,\mathrm{m}$ (horizontal by vertical) correlation length scales.

The above description produces individual realizations of sections across the front. We also generate a composite section using the profiles from all ship and glider transects, to display the average hydrography and geostrophic currents across the PF. The composite section is defined as a 1 km by 1 m (horizontal by vertical) resolution grid along the ship transects across the front (Figure 2). Next, all hydrography data from the ship's CTD system and the CT sensors of the MSS and gliders were objectively mapped horizontally at each depth level onto the composite grid using a 30 km by 30 km correlation length scale. Similarly, depth-averaged currents from the SADCP and gliders were horizontally mapped onto the composite grid. Hydrography data from gliders and DAC from gliders and SADCP were bin-averaged in 3 km by 3 km horizontal bins before horizontally mapping. A final objective mapping of the hydrography data was performed along the composite grid using $L_x = 30\,\mathrm{km}$ and $L_z = 30\,\mathrm{m}$.

Because an objectively mapped field is not natural data, errors in $\Theta$ and $S_A$ can propagate into density calculations and result in spurious unstable layers. To avoid this, we calculated density and, if necessary, rearranged the values to ensure gravitationally stable profiles. To ensure physical consistency, $S_A$ was then reproduced from these sorted density fields and $\Theta$. Note, however, that the difference between the originally mapped $S_A$ fields and the reproduced fields is minor.

Geostrophic velocity fields relative to the sea surface were calculated using the dynamic height as the geostrophic stream function using the Gibbs seawater library (McDougall and Barker, 2011). We used the objectively mapped $\Theta$ and $S_A$ sections from the individual transects, as well as the composite sections to produce relative geostrophic velocity. Finally, we obtain the absolute geostrophic velocity, $u_g$, by removing the depth average relative geostrophic velocity and adding the observed depth-averaged currents from the SADCP or glider in that section.

### 3.3 Turbulence measurements

When processing the shear probe data to estimate dissipation rates, we followed the recommendations and conventions of the SCOR Working Group on "Analyzing ocean turbulence observations to quantify mixing" (ATOMIX, http://wiki.uib.no/atomix, Lueck et al., 2023, preprint).

The dissipation rate of turbulent kinetic energy, $\varepsilon$, was estimated from the spectral analysis of high-pass filtered and despiked time series from the shear probes. Shear spectra were estimated using record lengths of 10 s for the glider/MR and 6 s for the

MSS measurements. We used fast Fourier transformation (FFT) lengths of 2 s that are cosine windowed and overlapped by 50%. Record lengths for spectral calculations, hence dissipation estimates, were overlapped by 50% (i.e., dissipation estimates from the MR are at 5 s time resolution and from the MSS at 3 s). Shear spectra were converted from frequency, $f$, domain to wavenumber, $k$, domain using Taylor's frozen turbulence hypothesis and the instrument speed through the water, $U$, as $k = f/U$. Here, $U$ is the flow past sensor estimate for the glider and the smooth fall rate for the MSS. The shear spectrum signal coherent with the accelerometer spectrum signal was removed using the Goodman method (Goodman et al., 2006).

Because all dissipation estimates were less than $10^{-5}$ W kg$^{-1}$, the rate of dissipation was estimated by spectral integration. For the glider, the measured shear components are $\partial v/\partial x$ and $\partial w/\partial x$, where $x$ is pointing along the axis of the instrument. For the MSS profiler, $\partial u/\partial z$ or $\partial v/\partial z$ are measured. Assuming isotropic turbulence, $\varepsilon$ was calculated for probe by integrating the cleaned wavenumber spectrum, $\Psi(k)$:

$$\epsilon = \frac{15}{2}\nu \overline{\left(\frac{\partial v}{\partial x}\right)^2} = \frac{15}{2}\nu \int_0^\infty \Psi(k)dk \approx \frac{15}{2}\nu \int_{k_1}^{k_u} \Psi(k)dk \tag{1}$$

where $\nu$ is the temperature-dependent kinematic viscosity, the overbar denotes averaging in time, and an arbitrary shear component is exemplified (e.g., Fer et al., 2014). The lower ($k_1$) integration limit is determined by the wavenumber corresponding to the FFT length, and the upper ($k_u < \infty$) integration limit is determined from a minimum in a low-order polynomial fit to the wavenumber spectrum in log-log space. Electronic noise typically takes over after the upper limit, and to account for the variance in the unresolved part of the spectrum the model spectrum of Lueck (2022b) was used. This is similar to the empirical Nasmyth spectrum (Nasmyth, 1970), but is a better-constrained approximation based on more than 14,000 shear spectra. Final dissipation estimates were obtained after quality screening following the ATOMIX recommendations, and averaging the good values from both probes or using the only good estimate. When the fraction of data removed by the despiking algorithm was greater than 5%, or when the figure-of-merit (a measure of misfit to the model spectrum) relative to the Lueck spectrum was greater than 1.15, or when the difference between dissipation estimates from two probes exceeded the expected statistical uncertainty at 95% confidence level (Lueck, 2022a), the estimate was flagged as bad. In addition, MSS dissipation measurements obtained from the ship were flagged in the upper 10 m because of the disturbance from the ship's draft, and the profiler's adjustment to free fall. Noise level of the dissipation rate measured by the MSS and the MR was about $(1-3) \times 10^{-9}$ and $(1-5) \times 10^{-11}$ W kg$^{-1}$, respectively.

### 3.4 Eddy kinetic energy

Surface geostrophic velocity anomalies from sea level anomalies obtained from satellite altimeter products (see Section 2.6) are used to calculate the surface eddy kinetic energy (EKE):

$$\text{EKE} = 0.5 \times (u_{ga}^2 + v_{ga}^2), \tag{2}$$

where $u_{ga}$ and $v_{ga}$ are the two components of the geostrophic velocity anomalies. For the region over the sill, enclosed by the yellow square in Figure 1b, we produce a spatially averaged EKE time series covering the duration of our data collection

(Sep 2019 to Mar 2021). For comparison, an EKE climatology is computed by averaging the monthly EKE over decades in the same region. Finally, we also compute the temporal average EKE at each grid point, averaged over the duration of our data collection. In all EKE estimates, SLA measurements where SIC was above 15% have been discarded. Hence the average EKE presented here is likely not representative of the region in periods when the region is mainly covered by ice. EKE is known to be stronger when SIC is low compared to times when SIC is high (von Appen et al., 2022).

## 4   Results

We present data from two scientific cruises conducted in the Barents Sea during October 2020 and February 2021 (referred to as the fall cruise and the winter cruise). The data are supplemented by data from four glider missions across the Barents Sea PF between the Spitsbergen Bank and the Great Bank in the time period 2019–2021.

### 4.1   Sea surface and atmospheric conditions during the cruises

During the fall cruise, the northwestern Barents Sea was completely free of sea ice, and sea surface temperature (SST) exceeded 0°C (Figure 1b). Only near the eastern coast of Svalbard and in the northernmost region, in the vicinity of the sea ice edge, SST decreased below 0°C. This is in stark contrast to the winter conditions where most of the northwestern Barents Sea was covered by sea ice, and the 0°C surface isotherm closely followed the 200 m isobath (Figure 1c). The surface signature of the AW inflow is confined to the waters where the seafloor depth exceeds 200 m, both during fall and winter.

Our study region is centered at the topographic sill between the Hopen trench and Olga Basin, and covers the PF region where AW from the south flows beneath PW to the north (yellow square, Figure 1b and c). Changing winds shift surface waters, affecting the surface temperature and sea ice concentration (Figure 1d and e). Most pronounced are the wind events in the beginning of February 2020 and late February 2021, when southerly winds changed the average sea ice concentration over the sill from nearly 100 % to less than 15 % over the course of a week. In addition, southerly winds lasting for nearly a month and a half in December 2020 and January 2021 kept the region above the sill nearly ice free until February 2021. During the previous winter, however, the sea ice concentration in the region above the sill rose gradually from early December 2019 to mid-February 2020.

While the surface conditions in the study region are highly influenced by the local surface winds, the properties and variability of the deeper layers are more resilient to the rapid shifting winds, and may be affected by other forcing mechanisms. Next, we look at the vertical and latitudinal structure of the PF across the sill, focusing on the oceanic drivers of variability.

### 4.2   Polar Front structure and seasonal variability

A total of 16 ship and glider transects were conducted during late summer, fall and winter 2019-2021. 11 of these were conducted during October and November and have been combined into a fall composite, while 4 of these were collected during December and February and have been combined into a winter composite (Figure 3a and b, respectively). Note that one transect conducted during August was not included in the seasonal composites. The method for combining data from individual

transects into a composite section is described in Section 3.2. During fall, a 50 m thick surface layer with warm ($> 1.5°$C) and

fresh ($< 34.6\,\mathrm{g\,kg^{-1}}$) water is present, and the PF is located below the surface layer. The center of the PF is highlighted by

the 0°C isotherm, a threshold practically identical to 0.1°C determined by Kolås (2024, Ch. 3.2.7). The PF, roughly 30 km

wide during fall (-1°C $< \Theta <$ 1.5°C at about 100 m depth), separates a subsurface AW core on the southern side of the sill

from PW on the northern side (Figure 3a). The subducted AW core is separated from the surface layer by a colder, 25 m thick,

interleaving layer at about 60 m depth. The overall PF structure during fall is similar to that from ship and glider transects

collected during October and early November 2020 (Figure 4 and 5). Note, however, that the glider transects in general exhibit

more variability in the PF structure than the ship transects. The short-term variability is described in Section 4.3.

From fall to winter, waters on both sides of the front cool, and the warm and fresh surface layer transitions into PW becoming

colder and more saline (Figure 3b). During winter, PW reaches about 60 km farther south than during fall, extending to about

330 -55 km relative to the saddle point of the sill. On the other hand, mAW and wPW below 125 m depth extend 5-10 km farther

north during winter than during fall, reaching about 25-30 km north of the sill. By combining data from all the individual

transects we produce an average composite section (Figure 3c). The composite section shows that, on average, the front is

located between 10 km south and 25 km north of the sill, starting at 50 m depth on the southern side and extending to the

seabed on the northern side. Note, however, that the bulk of our data is collected during fall and the composite is biased

towards "fall conditions". The across-front gradients of $\Theta$ and $S_A$ calculated from the average composite section, between

$\pm 20$ km of the 0°C isotherm at 100 m depth, are -0.06°C km$^{-1}$ and -0.008 g kg$^{-1}$ km$^{-1}$, respectively (Figure 3 and Table 4).

The 100 m depth was chosen as it is typically where the subsurface warm core is located south of the front. Averaging in the

composite section smooths the front, and the mean values of the gradients calculated from the individual transects are larger:

-0.09 $\pm$ 0.03 °C km$^{-1}$ and -0.010 $\pm$ 0.003 g kg$^{-1}$ km$^{-1}$ ($\pm$ one standard deviation).

Throughout most transects, the horizontal temperature and salinity gradients below the surface layer result in a baroclinic

front with density lines sloping downward as they cross the front. This is captured in the composite sections (Figure 3). On

the warm side of the front, during fall, water with $\sigma_0 = 27.8\,\mathrm{kg\,m^{-3}}$ resides about 70 m higher up in the water column than

on the cold side of the front (Figure 3a). Subsequently, a bottom-intensified, eastward geostrophic current is generated with

an estimated volume transport of 0.2 Sv. During winter, the frontal region extends about 50 km farther south. The 27.8 kg m$^{-3}$

isopycnal outcrops at -55 km, suggesting water with equal density resides 150 m higher up in the water column south of the

front compared to north of the front (Figure 3b). Nevertheless, the slope of the isopycnal across the front and the eastward

volume transport during winter are similar to those during fall (Table 4). Furthermore, the position of the frontal current is

also similar during fall and winter, with the core of the current located about 10 km north of the sill. The average transport by

the frontal current, calculated as the average from the individual transects, is $0.2 \pm 0.6$ Sv, where $\pm 0.6$ denotes the standard

deviation over the individual transects. The variability is relatively large, ranging between 1.1 Sv and -0.8 Sv. Negative volume

transports are due to reversals of the geostrophic current (Figure 4c and 5c), which are discussed in Section 4.3. If we exclude

the reversal events (3 transects), the mean eastward volume transport becomes $0.4 \pm 0.4$ Sv.

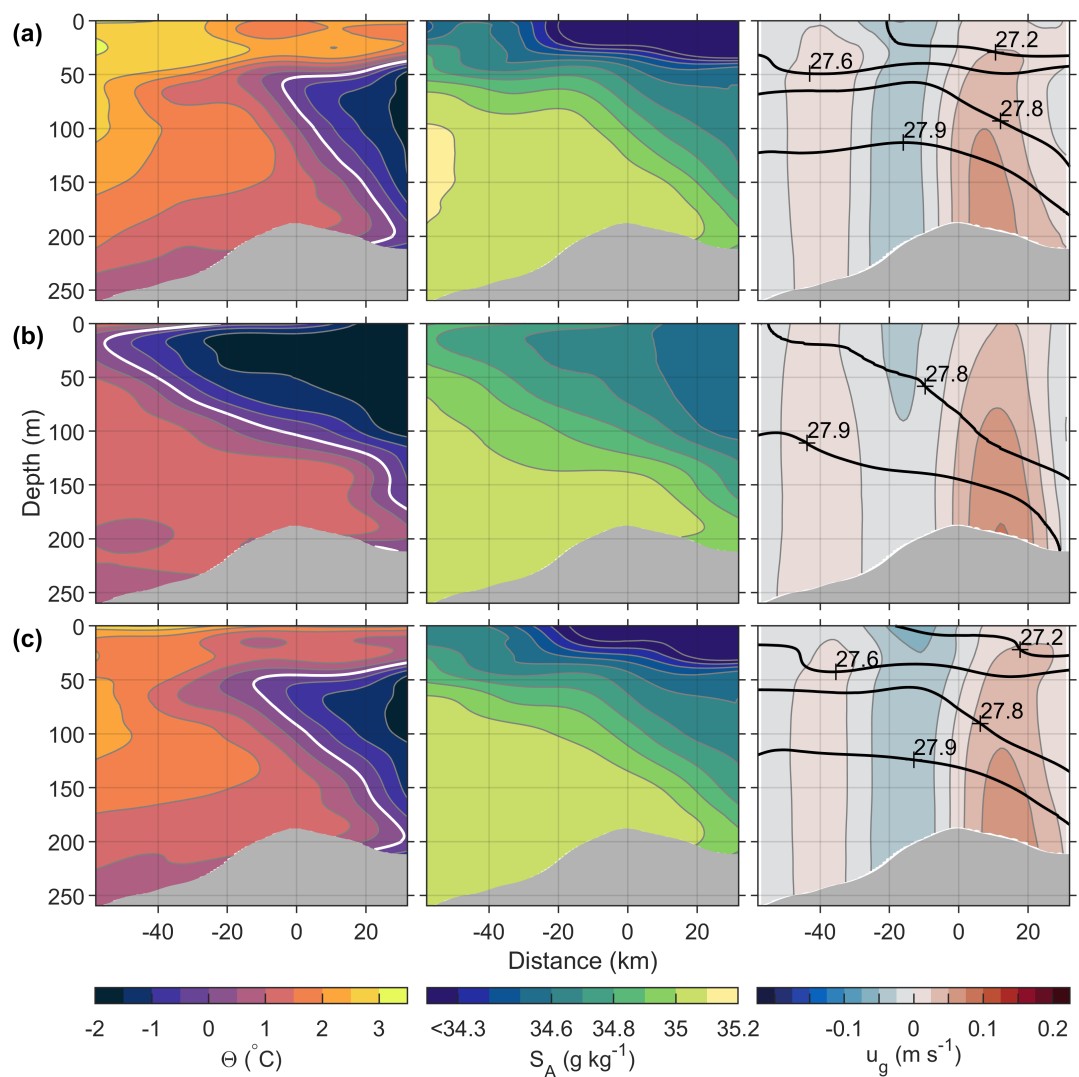

**Figure 3.** Composite section combining hydrography and current data from the individual ship and glider transects. **(a)** Fall composite combining data collected during October and November (11 transects), **(b)** winter composite combining data collected during December and February (4 transects), and **(c)** average composite combining all the individual transects (16 transects when including the August transect). From left to right, the different columns show Conservative Temperature, $\Theta$, Absolute Salinity, $S_A$, and absolute geostrophic velocity, $u_g$. Positive values of $u_g$ are eastward. The white line (left) is the 0°C isotherm, indicating the center of the Polar Front when below 50 m depth. Black contours (right) show isopycnals. The horizontal axis is the northward distance relative to the crest of the sill separating Hopen Trench in the south from Olga Basin in the north.

**Table 4.** Transport estimates for the frontal jet and gradients across the front for the composite sections and the individual transects shown in Figures 3, 4 and 5. Dates correspond to the dates of the individual transects as shown in Figures 4 and 5. The fall composite combine data collected during October and November (11 transects), the winter composite combine data collected during December and February (4 transects), and the average composite combine all the individual transects (16 transects). Volume transport estimates (positive eastward) are given for the frontal jet where it is captured. Gradients are calculated between $\pm 20$ km of the $0^\circ$C isotherm at 100 m depth. In the few cases where the transect does not extend $\pm 20$ km of the $0^\circ$C isotherm, the gradient is calculated from the edge of the transect. Mean and standard deviation ($\pm$ std) values are calculated from the individual transects.

| Date | Volume transport (Sv) | Temperature gradient ($^\circ$C km$^{-1}$) | Salinity gradient (g kg$^{-1}$ km$^{-1}$) | Density gradient (kg m$^{-3}$ km$^{-1}$) |
|---|---|---|---|---|
| Fall composite | 0.20 | -0.08 | -0.009 | -0.003 |
| Winter composite | 0.22 | -0.04 | -0.006 | -0.003 |
| Average composite | 0.19 | -0.07 | -0.008 | -0.003 |
| 14-Oct-20 17:41 | 0.7 | -0.10 | -0.012 | -0.004 |
| 15-Oct-20 05:41 | 0.5 | -0.10 | -0.012 | -0.004 |
| 15-Oct-20 11:23 | 0.5 | -0.09 | -0.010 | -0.003 |
| 15-Oct-20 17:56 | 0.1 | -0.09 | -0.011 | -0.003 |
| 17-Oct-20 22:08 | -0.8 | -0.08 | -0.009 | -0.003 |
| 15-Feb-21 18:22 | 0.1 | -0.04 | -0.007 | -0.003 |
| 16-Feb-21 06:49 | 0.3 | -0.04 | -0.006 | -0.003 |
| 18-Aug-19 | 0.9 | -0.08 | -0.007 | >-0.001 |
| 15-Nov-19 | - | -0.10 | -0.006 | -0.001 |
| 10-Dec-19 | 0.2 | -0.05 | -0.004 | -0.001 |
| 13-Dec-19 | - | - | - | - |
| 12-Oct-20 | 0.2 | -0.08 | -0.009 | -0.002 |
| 19-Oct-20 | -0.5 | -0.09 | -0.009 | -0.002 |
| 22-Oct-20 | 0.1 | -0.12 | -0.014 | -0.004 |
| 6-Nov-20 | 1.1 | -0.10 | -0.013 | -0.005 |
| 15-Nov-20 | -0.7 | -0.13 | -0.015 | -0.003 |
| Mean $\pm$ std | $0.2 \pm 0.6$ | $-0.09 \pm 0.03$ | $-0.010 \pm 0.003$ | $-0.003 \pm 0.001$ |

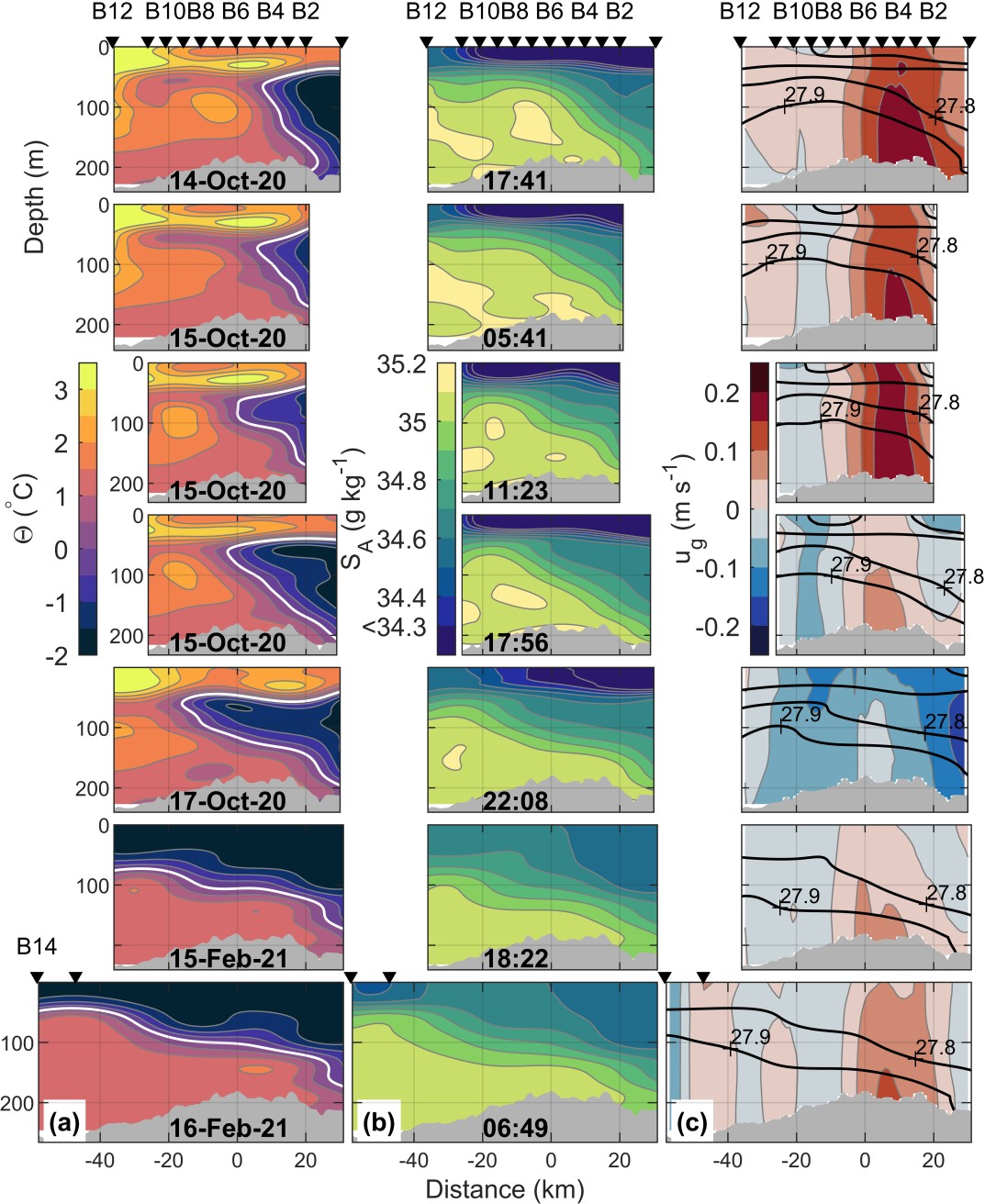

**Figure 4.** Repeated hydrographic measurements from the ship, across the Polar Front and the sill separating the Hopen Trench and the Olga Basin (Figure 1 and 2). **(a)** Conservative Temperature, $\Theta$, **(b)** Absolute Salinity, $S_A$, and **(c)** absolute geostrophic velocity, $u_g$ (positive eastward). The white line in (a) is the $0°C$ isotherm, indicating the center of the Polar Front when below $50\,m$ depth. Black contours in (c) show isopycnals at 27, 27.4 27.8 and $27.9\,kg\,m^{-3}$. The date and time (UTC) of the station at the crest of the sill are given in (a) and (b), respectively. The horizontal axis is northward distance relative to the crest of the sill ($0\,km$), and black triangles at the top indicate station locations with names. Bottom topography (gray) is from the ship's echosounder. The vertical-axis is scaled so that the height of $100\,m$ depth is equal in all panels.

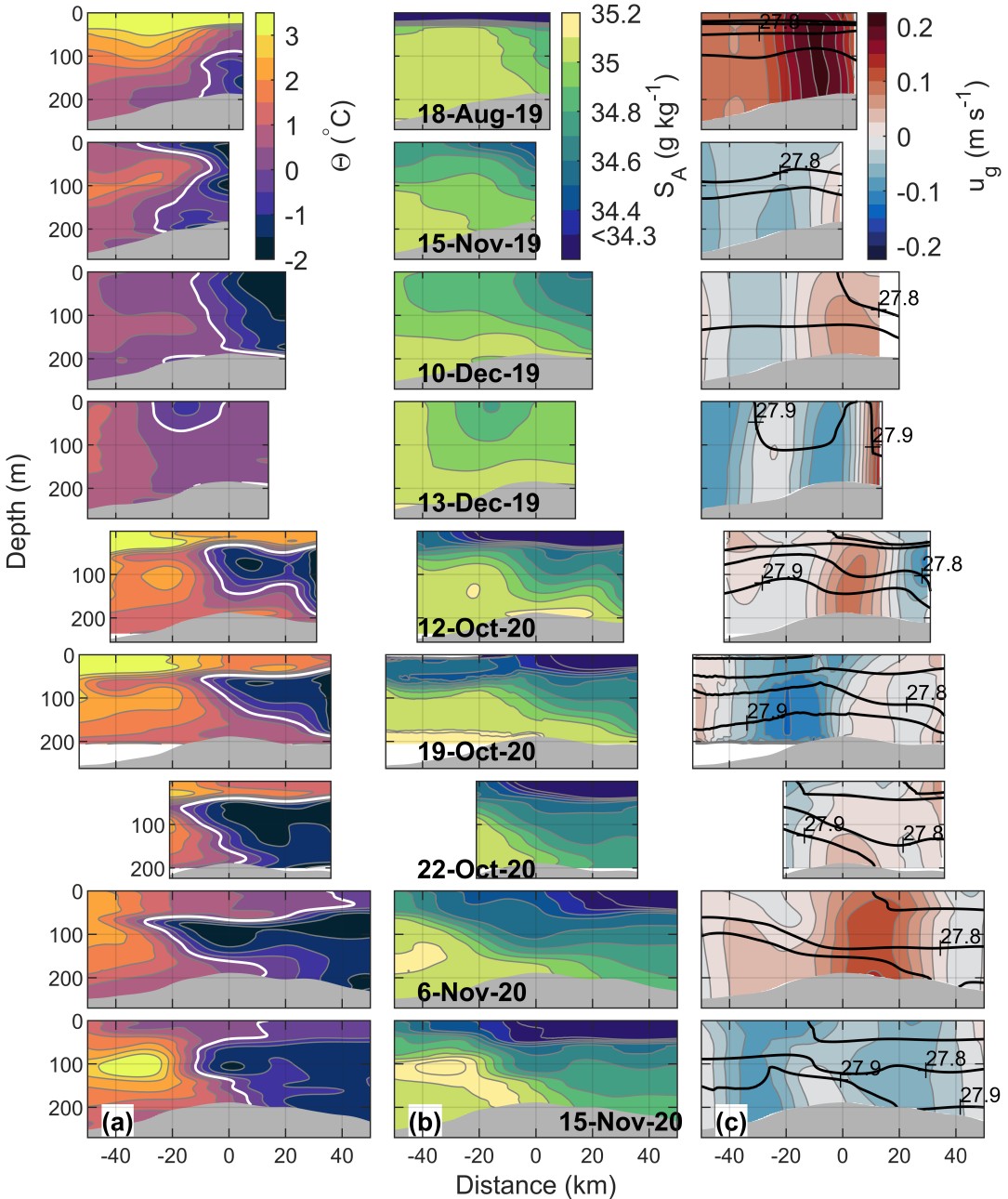

**Figure 5.** Same as Fig. 4 but for repeated glider transects across the sill. The date when the glider crosses the highest point on the sill is given in (b). Bottom topography (gray) is from IBCAO-v4.

### 4.3 Short-term variability at the Polar Front

Between 15 and 17 October 2020 the eastward geostrophic current on the sill weakened, and reversed, flowing westward (Figures 4 and 5). Simultaneously, the maximum southward extension of the PF increased by more than 25 km. We propose that this change was caused by an anticyclonic eddy in the Olga Basin north of the sill.

Satellite observations of SLA suggest an anticyclonic eddy developed after October 12, north of the sill (Figure 6a). This eddy moved south while it picked up in strength reaching a maximum on 19 October, before leaving the site after a few days. The geostrophic velocity anomalies calculated from SLA agree well with the glider DAC (Figure 6a). Note, however, that the DAC from the glider also measures the frontal current, hence the glider DAC and the surface geostrophic velocity anomalies calculated from SLA are not expected to be equal.

The surface signature of the eddy (anomalously high positive sea level) is supplemented by the subsurface data from the glider (Figure 6b and c). The eddy is a cold-core eddy, and as it moves south towards the PF, warmer water from above is pulled down and warmer and saltier water from below is lifted up (yellow boxes, Figure 6b–c). Subsequently, as the eddy moves across the front, the steepness of the isopycnal slope has reduced, suggesting the eddy consumed some of the available potential energy in the baroclinic front (middle panel, Figure 6c).

The anticyclonic eddy traversing the PF during mid-October 2020 is not a unique occurrence. Between 6 and 15 November 2020, a warm-core anticyclonic eddy moved northward, up the Hopen Trench (Figure 5, lower two panels). This eddy, also supported by satellite SLA observations (not shown here), transports warmer AW from further south towards the front, resulting in increased temperatures on both sides of the front, and a relaxation of the isopycnal slope.

While accounting for much of the variability observed across the PF, eddies do not explain all of it. The glider transects conducted between 10 and 13 December 2019 suggest the PF has shifted 20–40 km northward in the span of three days (Figure 5). This is likely not the case. The transect on 13 December is the easternmost transect across the sill, about 25 km east of 30°E (see glider sections shown in Figure 2) and does not extend as far north relative to the highest point on the sill as most of the other transects. Hence it is likely that the transect failed to capture the front as a result of spatial variability but not temporal variability.

Another source for temporal variability at the PF is tidal currents. Stations occupied for 12 hours with half-hourly repeated MSS casts conducted during both cruises showed that tidal currents dominated short-term current variability at the PF during the fall and winter cruises (Figure 7a,d). The observed depth-averaged currents reach a maximum northward velocity of ∼15 cm s$^{-1}$ during both fall and winter. While the following southward flow is similarly strong during fall, it only reaches about 5 cm s$^{-1}$ during winter. Note that on 18 October, the eddy displayed in Figure 6a is close to the repeated station. The azimuthal velocity of the rim of the eddy, with a west-southwestward component, may be responsible for the systematic deviation of DAC from the Arc2km tidal currents (Figure 7a). During fall, maximum northward velocities are accompanied by substantial warming in the middle of the water column (Figure 7b). This may be interpreted as a northward displacement of the front, allowing warmer AW to reach the measurement station throughout the water column. During winter, this is not visible, possibly due to the nearly horizontal nature of the front (lower two panels of Figure 4). The suspected lateral displacement of the front

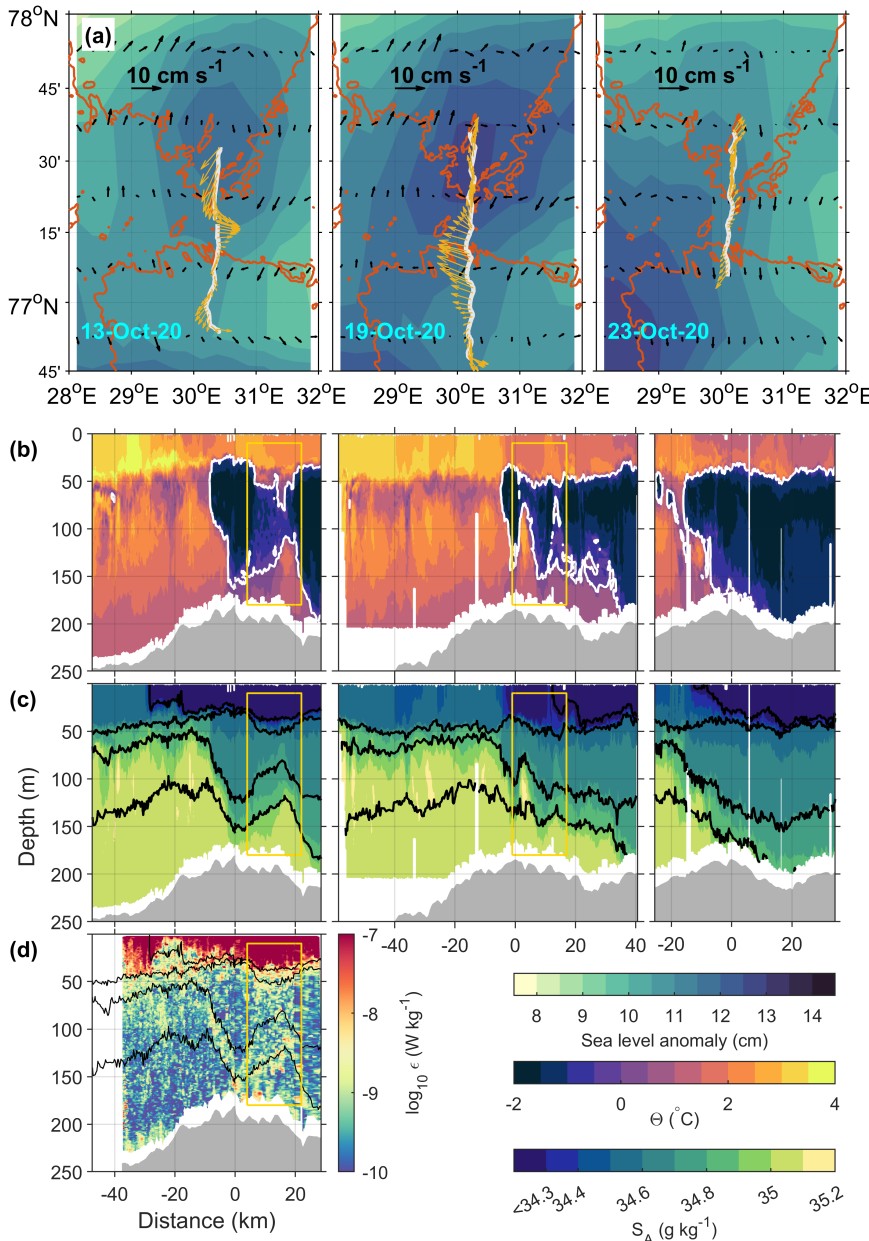

**Figure 6.** Three crossings over sill with the Slocum glider Odin in October 2020. **(a)** Sea level anomalies and the surface geostrophic velocity anomalies (black quivers) on the date specified in the lower left corner. White line shows glider track. Yellow quivers show glider DAC. **(b)** Conservative Temperature, $\Theta$, and **(c)** Absolute Salinity, $S_A$, along the three glider crossings shown in (a). **(d)** Dissipation rate of turbulent kinetic energy, $\varepsilon$, along the first glider transect shown in (a) (turbulence instrument stopped recording after the first transect). White line in (b) is the 0°C isotherm and black lines in (c) and (d) are the 27.2, 27.6, 27.8 and 27.9 kg m$^{-3}$ isopycnals. The glider transects shown in (b) and (c) are the same as the October transects in Figure 5, but without objective mapping or along path smoothing. The yellow boxes enclose the position of, and region most affected by, the eddy. Bottom topography (gray) is from IBCAO-v4.

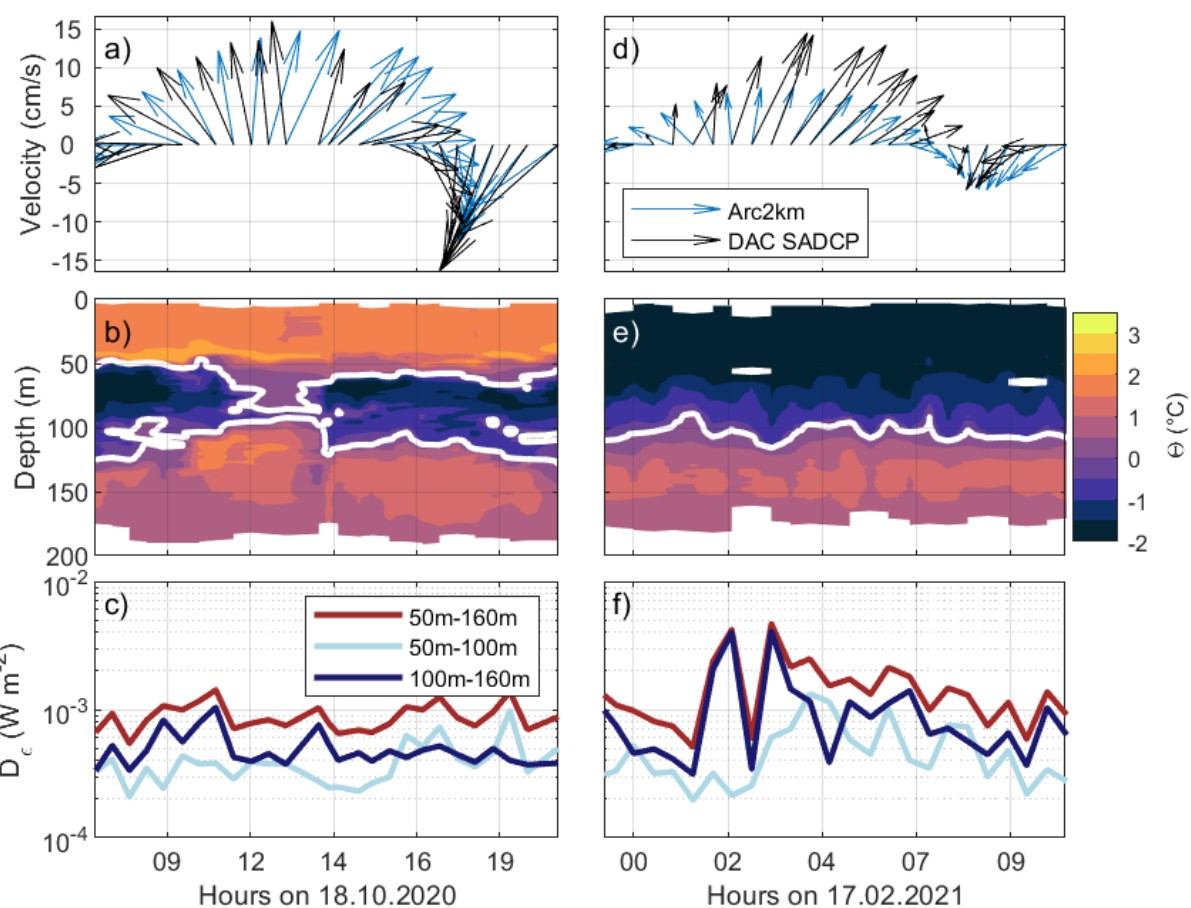

**Figure 7.** 12-hour repeat stations at B5 in fall (left) and B7 in winter (right). Total depth is 192 m and 185 m for B5 and B7, respectively, and the station location is indicated in Figure 2. (**a,d**) Time-series of depth-averaged currents (DAC) together with tidal currents from the Arc2km model (Howard and Padman, 2021), (**b,e**) temperature with white line marking the 0°C isotherm, and (**c,f**) dissipation rate vertically integrated over different layers $D_\epsilon$ indicated in the legend. Maximum layer depth is set to 160 m to be consistent with the few profiles that do not extend deeper.

during the fall cruise does not coincide with any significant changes in the layer-integrated dissipation rates (Figure 7c). However, during winter, peak tidal currents are associated with an increased dissipation rate in the deeper part of the water column (Figure 7f). The vertically-integrated dissipation rate, $D_\epsilon = \rho \int_{z_1}^{z_2} \epsilon\, dz$, where $\rho = 1025\,\mathrm{kg\,m^{-3}}$, and $z_1$ and $z_2$ are the
bounds of the layers over which integration is made, indicates the total dissipation rate per unit surface area, and can be related to work done, for example, by tidal currents. In winter, $D_\epsilon$ in the deeper part of the water column accounted for a substantial fraction of the dissipation in the water column below the surface layer.

The tidal $v$-component is likely to have a greater impact on the frontal structure than the $u$-component as the front is aligned west-east. However, the relatively weak north-south tidal oscillation is unlikely to shift the water any more than about 1 km

within a tidal period, with minimal effect on the frontal structure and geostrophic balance. Nevertheless, tidal-induced velocity
shear may generate mixing which again may erode the frontal structure.

## 4.4    Mixing across the Polar Front

Microstructure measurement transects were conducted from the ship across the front in the fall and winter cruises, and also
by the glider in October 2020. Most of the observed mixing occurs within the surface boundary layer, where estimates of
dissipation rate of turbulent kinetic energy ($\varepsilon$) exceed $10^{-6}\,\mathrm{W\,kg^{-1}}$ (Figure 6d and 8). In October, the surface mixed layer
(upper $60\,\mathrm{m}$ or so) is separated from the deeper layer by a strongly stratified pycnocline where mixing quickly abates. In
February however, the stratification in the pycnocline is weaker, and surface mixing occasionally extends down to $100\,\mathrm{m}$
depth. Below the surface mixed layer, most of the mixing occurs in the bottom boundary layer where $\varepsilon$ reaches $10^{-7}\,\mathrm{W\,kg^{-1}}$
(Figure 6d and 8). Mixing in the bottom boundary layer is strongest during the first October ship transects when the geostrophic
current is at the strongest (Figure 8, upper two panels). At the front, where $\Theta$ and $S_A$ gradients are strong, we observe elevated
dissipation rates occasionally reaching $10^{-7}\,\mathrm{W\,kg^{-1}}$. This is particularly visible during the glider transect where we have
microstructure profiles at every $500\,\mathrm{m}$ interval across the front. Here, elevated dissipation rates follow the $0°\mathrm{C}$ isotherm closely,
marking the transition and mixing between mAW and PW (Figure 6b and d). In the interior, away from the front, estimates of
$\varepsilon$ are often at the noise level of the instrument, indicating little or no mixing.

The role of the PF as a mixing zone is highlighted by the observed water mass transformation across the front, on $\Theta$-$S_A$
space (Figure 9). Profiles from the front region are shown together with those from Section D in the south, and Section A
and F in the north (see Section locations in Figure 2). About $60\,\mathrm{km}$ south of the highest point on the sill, the water column
is composed of AW and mAW (Figure 9, Section D). These water masses have mainly been modified through cooling by
the atmosphere during their transit from the BSO toward the Hopen Trench. However, as the AW enters the frontal region,
AW mixes directly with PW residing north of the PF in the Olga Basin, generating wPW (Figure 9 and Figure 2). The wPW
is transported eastward by the geostrophic current along the front, and is found downstream northeast of the sill (Figure 9,
Section A).

## 5    Discussion

### 5.1    Barents Sea Polar Front structure and seasonal change

Previous studies on the western Barents Sea PF have been conducted along the southern and southeastern slopes of the Spitsber-
gen Bank (Loeng, 1991; Gawarkiewicz and Plueddemann, 1995; Parsons et al., 1996; Harris et al., 1998; Fer and Drinkwater,
2014), and along the southwestern slope of the Great Bank (Våge et al., 2014). These studies describe the PF in the west-
ern Barents Sea, below the surface layer, as a barotropic front with horizontal density lines. The PF we observe across the
sill between Spitsbergen Bank and Great Bank differs from these observations. We observe a baroclinic front where isopyc-
nals tilt downwards with distance from south to north across the front (Figure 3,4 and 5). Subsequently, a bottom-intensified

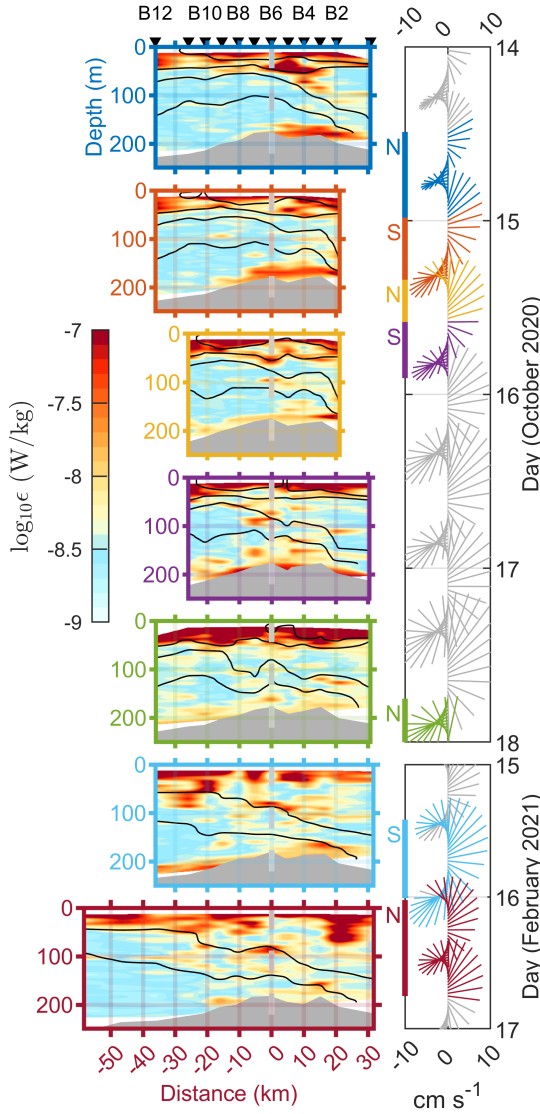

**Figure 8.** Dissipation rate of turbulent kinetic energy, $\varepsilon$, estimated from the MSS profiler during the fall (upper 5 panels) and winter (lower two panels) cruises. Black lines show isopycnals at 27.2, 27.6, 27.8 and 27.9 kg m$^{-3}$. Tidal currents during the transects are shown in the right panels. Colors indicate the individual transects with S or N indicating whether the section started from the south or north end, respectively.

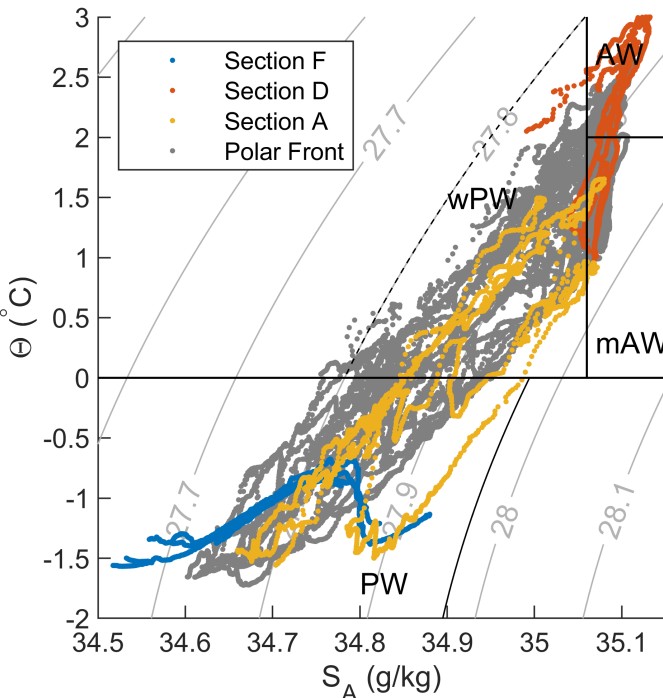

**Figure 9.** $\Theta$-$S_A$ diagram showing water masses around and across the Polar Front, below 80 m depth. Blue, yellow and red profiles correspond to the profiles along the indicated sections, marked with the same color in Figure 2. Polar Front profiles (gray) are profiles collected between stations B2 and B11. All profiles are from the fall cruise.

geostrophic flow develops. A likely reason for this difference in the frontal structure is the difference in the bathymetry. While the seabed on the Spitsbergen Bank and the Great Bank rises to above 100 m depth, the highest point on the sill is 180 m deep. Along the slopes on the Banks the seabed serves as a boundary, blocking the AW. Across the sill, AW can flow without the same topographic boundary and consequently a geostrophic adjustment balances the inflowing AW and the overlaying PW.

The upper 50–80 m of the water column across the topographic sill is largely affected by seasonal variability (Figure 3a and b). Through August to early November, relatively warm ($> 2°C$) and fresh water ($< 34.3\,\mathrm{g\,kg^{-1}}$) resides in the surface layer, suggesting atmospheric heating of seasonal melt water (Figure 4 and 5). By February, nearly all of the surface layer is occupied by PW ($< 0°C$), and the salinity has increased ($> 34.6\,\mathrm{g\,kg^{-1}}$), reducing the strong stratification in the pycnocline observed during fall and extending the PF more or less to the surface. The increased surface salinity during winter is likely due to brine

rejection from sea ice formation. This seasonality is similar to the seasonality previously described along the southern slope of the Spitsbergen Bank (Harris et al., 1998).

In contrast to the seasonal change observed in the upper 100 m, the position of the front below 100 m is relatively stable from fall to winter (Figures 4). However, the position of the incropping of the 0°C isotherm on the north side of the sill varies by about 10 km, suggesting that the AW overflow across the sill below 100 m depth exhibits some seasonal variability. The

temperature and salinity in the bottom 50 m above the sill remain above 1°C and 35 g kg$^{-1}$ between October and February,

suggesting little seasonal change. Nevertheless, there is more mAW and wPW in the bottom 50 m, north of the sill, during winter compared to fall. This suggests a larger influence of warm water in the bottom layer during winter than during fall, although temperatures south of the front in general are lower during winter compared to fall.

The difference in seasonality between the upper and lower layer (above and below 100 m depth) has consequences for the PF structure. During fall, the 0°C isotherm, particularly between 50–100 m depth, is typically inclined from the horizontal and is occasionally vertical, while it is nearly horizontal during winter. Oziel et al. (2016) describe the PF position as the isotherm corresponding to the modal temperature in the region with the highest horizontal temperature gradients in the 50–100 m layer. While this may be representative of the PF location during fall, it is less suitable during winter as the temperature front becomes nearly horizontal, spanning almost 100 km. Nevertheless, the density gradient across the front at 100 m depth remains similar from fall to winter, and is on average -0.003 $\pm$ 0.001 $\mathrm{kg\,m^{-3}\,km^{-1}}$ (Figure 3 and Table 4). This is about one-tenth of the density gradient across the surface density front observed by Parsons et al. (1996). However, the surface and subsurface fronts are two different domains and are not directly comparable. Most frontal studies in the Barents Sea have studied the subsurface temperature and salinity gradients individually as they tend to be density compensating (Oziel et al., 2016; Barton et al., 2018). However, estimated from Figure 7 in Barton et al. (2018), the 1985-2016 average density gradient across the Ludlov Saddle in the Barents Sea, at 100m depth, is approximately -0.003 $\mathrm{kg\,m^{-3}\,km^{-1}}$, similar to our findings.

## 5.2 Short term variability

The middle layer of the water column above the sill (roughly between 50–130 m depth) is highly dynamic, particularly during fall and early winter when the temperature and salinity fronts are more vertical than during winter (Figures 4 and 5). In October we observe the position of the front, marked by the 0°C isotherm, shifting nearly 25 km over the span of 3 to 4 days (Figure 4). The two main drivers of this short term variability (below 50 m depth) are tidal currents and mesoscale eddies.

The tidal currents observed above the sill have a peak-to-peak variability of about 30 $\mathrm{cm\,s^{-1}}$ during fall, and 20 $\mathrm{cm\,s^{-1}}$ during February (Figure 7a and d). During fall, when the temperature front is nearly vertical, tidal currents shift warm and cold water back and forth. The north-south displacement of water masses is roughly 2 km when assuming an average tidal current of 10 $\mathrm{cm\,s^{-1}}$ during 6 hours. Nevertheless, the tidal-induced mixing is mainly confined to the bottom 30 m with little contribution to mixing at mid-depth, both during fall and winter (Figures 7 and 8). For comparison, Fer and Drinkwater (2014) observed tidal currents with a peak-to-peak variability of about 50 $\mathrm{cm\,s^{-1}}$ on the southeastern slope of the Spitsbergen bank during May 2008. As a result, estimates of $\varepsilon$ exceeded $10^{-6}$ $\mathrm{W\,kg^{-1}}$ near the seabed, and the entire water column was well mixed where the bank was shallower than about 75 m. We expect the contribution from mixing due to tidal currents to have large spatial variability along the PF, depending much on the local topography. Note that we observe mixing rates in the surface layer (upper 50 m) that are much higher than mixing rates in the lower layer throughout all of our observations, mainly exceeding $10^{-7}$ $\mathrm{W\,kg^{-1}}$ (Figure 6d and 8). This is comparable to previous mixing rates observed on the sill between the Hopen Trench and the Olga Basin. Sundfjord et al. (2007) found an average dissipation rate of $4.7 \times 10^{-7}$ $\mathrm{W\,kg^{-1}}$ across the pycnocline in May 2005, and stated that the average dissipation rates within the pycnocline were at least 5 times larger than that below the pycnocline.

In October and November 2020, we observed two eddies modifying the structure of the PF across the sill between the Hopen Trench and the Olga Basin (Section 4.3 and Figures 5 and 6). Porter et al. (2020) similarly observed a surface eddy further south along the Hopen Trench during July 2017. This eddy originated from the Arctic region in the Barents Sea and traveled south along the Spitsbergen Bank (Porter et al., 2020). Eddy kinetic energy calculated from satellite-derived surface geostrophic velocity anomalies suggests the slope between the Spitsbergen Bank and the Hopen trench is a region where eddies commonly occur (Figure 10a). In addition, EKE averaged over the sill between the Hopen Trench and the Olga Basin suggests several eddies were present between August 2019 and April 2021 (Figure 10b). Note however that the eddy observed in mid-October 2020 (shown in Figure 6) does not appear in the time series of the average EKE over the sill, likely due to a mismatch between the averaging box and the position of the eddy. Hence, the time series is likely an underestimate of the eddy energetics during our study period. In addition, only the largest eddies will have a surface SLA signal detectable by satellites as the SLA resolution is 0.25°. Accordingly, as the PF across the sill is baroclinic, eddies too small to be detectable by satellite altimetry may shed from baroclinic instabilities, feeding off the available potential energy across the front. Atadzhanova et al. (2018) observed about 3000 eddy structures in the Barents Sea during June-October 2007 and 2011, of which 25% occurred in the region around the PF. These eddies were determined from radar images and had a mean diameter of about 4 km. Most eddies at that scale will likely not be detectable by sea-level height measurements. In addition, Porter et al. (2020) state that the eddy they detected in the southern Hopen Trench was thermally capped, that is the surface of the eddy had lost its cold core characteristics and was indistinguishable from surrounding waters using satellite imagery. This suggests even the number of eddies observed by Atadzhanova et al. (2018) could be an underestimate of the eddies present in the Barents Sea.

A comparison of the 10-year monthly-average EKE suggests that eddies in the northwestern Barents Sea are most energetic during winter, by a factor of two relative to summer (Figure 10b). This is in contrast to the observations by Atadzhanova et al. (2018) who find the peak eddy activity to be in July. Note, however, that SLA measurements where SIC was above 15% have been discarded (see Section 3.4). This could lead to a potential bias towards higher EKE during winter as we only include the most energetic regions (ice-free regions) during winter. Our 10-year monthly-average EKE is likely only representative of the region when most of the region enclosed in Figure 10a is ice-free.

Comparing the 2000-09 decade to the 2010-19 decade suggests the EKE in general has increased in the region. The reason for this increase in EKE is not clear. The volume transport through the BSO is closely linked to the North Atlantic Oscillation (NAO) and local winds (Heukamp et al., 2023). An increased volume flux into the Barents Sea caused by faster, more energetic currents could potentially lead to an increase in observed EKE in the Barents Sea. However, there is little evidence supporting any significant difference in the volume flux through the BSO between 2000 and 2019 (Heukamp et al., 2023; Årthun et al., 2019). We suggest the increase in EKE may be related to the observed increase in oceanic heat transport into the Barents Sea, and the subsequent decrease in sea ice extent, the so-called Atlantification of the Barents Sea (Skagseth et al., 2008; Årthun et al., 2012; Barton et al., 2018; Årthun et al., 2019).

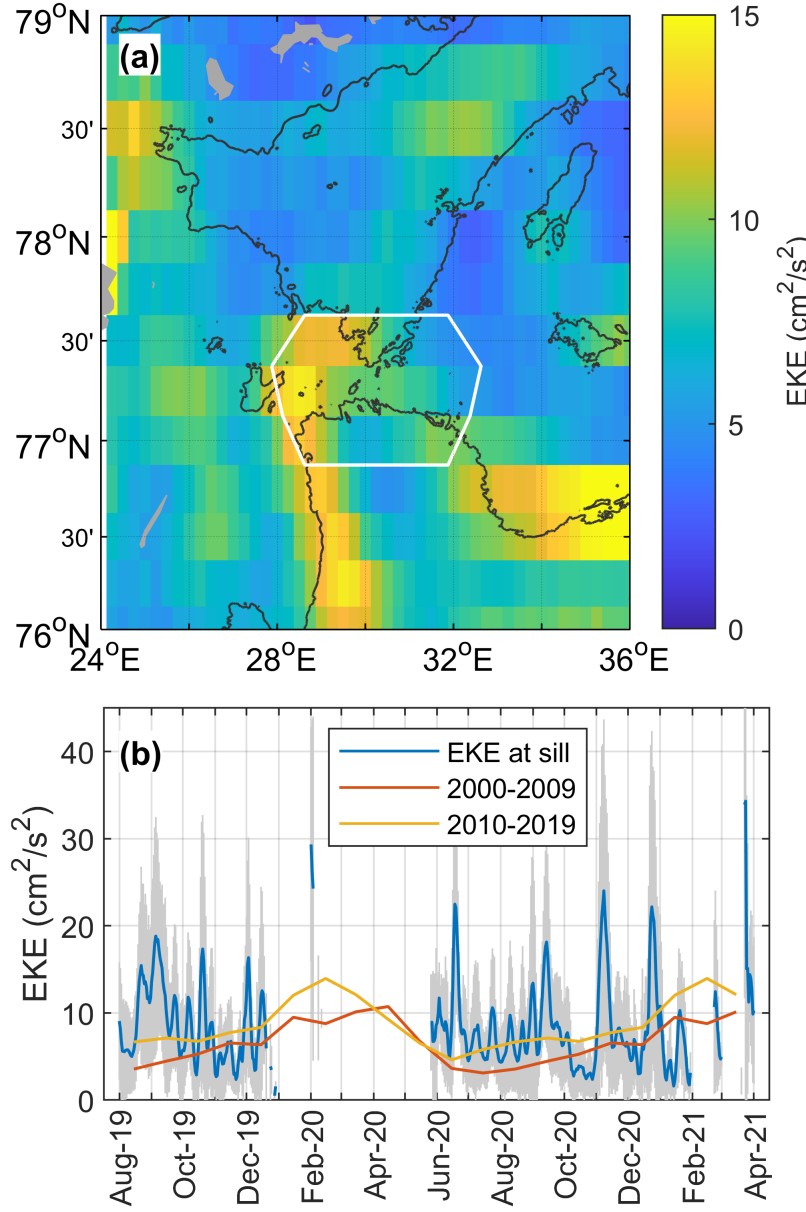

**Figure 10.** Eddy kinetic energy estimated from satellite-derived surface geostrophic velocity anomalies, based on sea level anomalies. **(a)** Temporal average between 1 August 2019 and 1 April 2021, covering the data collection period. Black line is the 200 m isobath. White line encloses the region used for spatial average in (b). **(b)** Blue line shows the spatially averaged daily EKE over the region enclosed by the white line in (a), for the duration of our data collection period. Gray shading shows the standard deviation of the averaged EKE. Red and yellow lines show the monthly decadal mean for the same region.

## 6 Summary

This study provides insights into the dynamics and hydrography of the Barents Sea Polar Front within the region bounded by the Hopen Trench and Olga Basin in the Barents Sea. The topographic sill separating the Hopen Trench from the Olga Basin is a location where AW meets PW, setting up a front where AW eventually subducts below the PW. The data presented herein was collected during two scientific cruises in October 2020 and February 2021, as well as four glider missions spanning 2019 to 2021.

During the fall cruise, the western Barents Sea was devoid of sea ice, with sea surface temperatures mainly exceeding $0°$C. In February however, the northwestern Barents Sea and the sill separating the Hopen Trench and the Olga Basin were covered by sea ice, with the $0°$C isotherm aligning closely with the 200m isobath. Despite the seasonal changes at the surface and in the surface mixed layer, the influence of the AW inflow near the seabed over the sill remained unperturbed.

This study shows that the PF over the sill is a distinct baroclinic front supporting an eastward geostrophic current above the sill. The average eastward transport of this current is estimated to $0.2 \pm 0.6$ Sv, where $\pm 0.6$ denotes the standard deviation over 16 repeat transects. The baroclinic front differs markedly from prior observations along the Spitsbergen and Great Banks slopes, where a barotropic front with horizontal density lines was typically noted. The distinction arises from variations in bathymetry, as the absence of a topographic boundary permits a geostrophic adjustment to balance inflowing AW and overlying PW.

The upper layers of the PF (0–100 m) experienced pronounced seasonal fluctuations, influenced by atmospheric heating, sea ice formation, and brine rejection. However, the position of the front beneath 100 m depth exhibited minimal seasonal variability.

Short-term variability stemmed from tidal currents and mesoscale eddies. Tidal currents induced 2–4 km north-south displacement of water masses, especially notable when the subsurface temperature front was approximately vertical. However, the modification of the PF by tidal currents was negligible compared to the effects of mesoscale eddies. Observations collected from ships and gliders as well as satellite altimetry data independently indicated the presence of two eddies at the PF in October and November 2020. These eddies influenced the front's structure, shifting the position of the front as much as 25 km in less than 4 days. In addition, the isopycnal slope across the front was markedly reduced after the passage of an eddy, suggesting the eddy reduced the available potential energy of the front during its traversing of the front. Simultaneously, as the eddy approached the front, satellite-derived SLA showed the eddy's radial velocity increasing, suggesting the available potential energy was converted to EKE.

Microstructure measurements showed intense mixing within the surface boundary layer, particularly in the upper 60 m. During winter, this mixing extended occasionally to 100 m. Below the surface layer, significant mixing was concentrated in the bottom boundary layer, and linked to the tidal oscillations across the sill. Nevertheless, we observed substantial water mass transformation across the front, which was likely a result of eddy-driven along isopycnal mixing.

This study offers a comprehensive description of the Barents Sea Polar Front, shedding light on its interactions with seasonal shifts, tidal currents, and mesoscale eddies. The distinctive baroclinic structure observed across the topographic sill underscores the importance of local bathymetry in shaping front dynamics.

*Data availability.* All data presented in this study are openly available. Hydrography, current and microstructure data collected during the October 2020 and February 2021 cruises are available from Fer et al. (2023a), Fer et al. (2023d), Fer et al. (2023e) and Fer et al. (2023b). Glider data are available from Kolås et al. (2022) and the microstructure data collected by *Odin* is available from Fer et al. (2023c). Sea ice concentration is from OSI-SAF (2017), sea surface temperature is from Copernicus (2019), daily and monthly sea level anomalies are from 545 Copernicus (2023), and bathymetry data are from (Jakobsson et al., 2020). The Arctic 2 km Tide Model (Arc2kmTM) can be downloaded from the Arctic Data Center at doi:10.18739/A2PV6B79W (Howard and Padman, 2021).

*Author contributions.* IF, TB and EHK collected the data in addition to conceiving and planning the analysis. EHK, IF and TB performed the analysis. EHK wrote the paper, with advice and critical feedback from IF and TB. All authors discussed the results and finalized the paper.

*Competing interests.* At least one of the (co-)authors is a member of the editorial board of Ocean Science. The authors have no other 550 competing interests to declare.

*Acknowledgements.* This work was supported by the Nansen Legacy Project, project number 276730. We thank the officers, crew and scientists of the RV G.O. Sars cruise in October 2020 and the RV Kronprins Haakon cruise in February 2021. We thank the glider team at NorGliders (http://norgliders.gfi.uib.no/), for operating and maintaining all the glider missions in challenging conditions in the Barents Sea.

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
