# Peer review of "The Polar Front in the northwestern Barents Sea: structure, variability and mixing"

_EGUsphere, 2023_

## Author Comment (AC3)

*Response to community comments on Kolås et al. EGUSPHERE-2023-2864, The Polar Front in the northwestern Barents Sea: structure, variability and mixing.*

We thank Dr. Maria Dolores Pérez-Hernández for the constructive comments and useful suggestions, which helped to improve the manuscript. Below we provide a point-by-point response to all comments. Community comments are reproduced in *italic type in red* followed by our response in regular type in black color. The future tense refers to our plan to address the comments when preparing the revised version.

**Response to community comments by Dr. Maria Dolores Pérez-Hernández**

*This study focuses on understanding the Polar Front over the sill between the Hopen Trench and Olga Basin, one of the four areas where AW meets Polar Water in the Barents Sea. The Polar Front is important for biological activity and mixing in the area. The results arise from two detailed fieldworks where hydrographic data from ship and glider sections are analyzed with altimetry, wind, and sea ice concentrations. This study is very interesting and highlights the high variability that the Polar Front has in terms of existing in location, shape, forcings, and time. The dataset used is available, and the study is relevant to the field. I suggest publishing it after some changes.*

Thank you for your valuable feedback!

*My main concern is Section 4.2, 'Polar Front structure and seasonal variability.' Here are some comments to help improve it:*

- *The way it is written reads more like a general variability than a seasonal cycle, and it finishes with an average view. Results will be better understood if the section starts with the average view and, from there, moves towards seasonality.*
  We agree. The section could benefit from a different structure, starting with the average/seasonal change and moving on to short-scale variability. We will improve on this.
- *The seasonal cycle cannot be fully resolved with the available dataset. Nevertheless, a section in August could be used as representative of summer, 11 sections can be combined into a fall average section, and 4 sections can be averaged as a winter section.*
  Thank you for suggesting this. We will attempt to divide the different synoptic sections into seasonal averages and discuss the results. However, the number of figures in the manuscript is already at a limit. We will add three new rows to figure 5 to show what you suggest.
- *This section also has some minor issues, like using the term 'Atlantic-origin water' instead of 'modified Atlantic Water' as stated in Table 3.*
  The reason for using the term "Atlantic-origin water" is that it includes all the water masses originating directly from the BSO and the Norwegian Atlantic slope current, not only the "modified Atlantic Water". That is, it is not only mAW reaching the front. No changes made.
- *It is not said whether negative distances are located north or south of the sill in the caption of Figure 3.*
  Indeed. We now insert a sentence clarifying this.

- *At some point, it is stated that the velocities from altimetry match the DAC, and while that is true for October 19, the agreement is not as evident on the other two dates.*
  We agree. The barotropic geostrophic velocity calculated from SLA is a good indicator of eddies, their presence and their strength. However, it may differ from the depth average current measured by the glider as the DAC also measures the frontal current. We will elaborate on this in the text.

- *Transport estimations are only given for the average section (Figure 5). You could also estimate seasonal transports or, if not, a table with the transport for each time frame.*
  We will add a table to include the transports from both the individual and seasonal transects.

- *Overall, the text between lines 306 and 317 should be carefully revisited as they had some misleading errors.*
  - *Line 306-307. AW is separated from the surface by a warm and fresh layer; it is not cooler but fresher in the upper 60m.*
    Indeed. What we tried to communicate was that the subsurface AW core is separated from the surface layer (which is warm and fresh) by a colder interleaving layer between the AW core and the surface layer. We now clarify this in the text.
  - *Line 312. I don't see a cooling from December to February, as the years don't match. I see cooling on the glider dataset for November and December 2019 and the ship dataset for February 2021. These colder sections are relative to the August 2019 and October 2020 sections.*
    We agree. We rewrote to clarify this.
  - *Line 314-315. In February, AW is not present (being AW defined with temperatures higher than 2ºC and salinities higher than 35.06). From what is visible in Figure 3 a, lower 2 subplots, the northern side of the Front fits better with the description of modified AW given in Table 3.*
    Indeed. Only mAW and wPW is present on the Atlantic side of the front, both a result of AW cooling and mixing with PW. This is why we name them Atlantic-origin water, because it is not AW but a product from AW. We rewrote these sentences to clarify.
  - *Line 317. Assuming that negative distances are south of the sill, the average position lies 10km north of the sill, where the core of the positive velocities is found (Figure 5). Yet, it can reach as far as 10 km south of the sill in the 50 m depth and narrows from there to the bottom.*
    We agree. This was our point, but we will further clarify it.

- *Lines 415-416 and 420. The seasonality of the isopycnals is arguable. It says that the isopycnal tilt is flat in winter, while in Figure 3, the February sections have quite a tilt. Although the Glider sections of December have flat isopycnals, some of the October sections also present nearly flat isopycnals. So, this goes again with Section 4.2; perhaps a seasonal composite could be a better approach to assess seasonality or just blend it all under 'variability'.*
  We agree that the seasonality of the isopycnals is difficult to assess based on the current figures. We now produce seasonal composites to better show the change.

*Some other minor issues:*

- *Line 86 to 87. This sentence is confusing; I suggest rephrasing or avoiding mentioning Figure 1b. Here, the text refers to the data used in the study, while Figure 1b introduces a larger area.*
  Agreed. We rewrote it as "An overview of the data coverage across and near the front is shown in Figure 2.".

- *Line 94. Please explain how salinity was calibrated (AUTOSAL, Portasal,other?)*
  Bottle samples are analyzed at IMR with a Guildline Portasal 8410 salinometer. Salinity and

conductivity values measured by the Portasal for each sample are compared with the corresponding CTD data. Following the procedure recommended by UNESCO, only data within the 95% confidence interval are used to correct the calibration of the CTD conductivity. We now add a sentence on this in the text:

"The CTD system was equipped with a SBE 32 Carousell fitted with bottles for collecting water samples at all stations. Bottle samples were analyzed using a Guildline Portasal 8410 salinometer and used to calibrate salinity."

- *Section 2.2. Two paragraphs above, it said that the cruises will be referred to as fall and winter cruises, but in this section, the names of the vessels are used. You could recall the season after the cruise name at the beginning or go with the seasonal names.*
  Agreed. We added the seasonal names.
- *Line 121. 'of the PF location (Figure 2)'.*
  Agreed.
- *Line 201. EUMETSAT OSI-SAF (2017).*
  Corrected.
- *Line 291. AW depth exceeds 200m depth? Do you mean that the entire water column is AW? or that it spreads to waters shallower than 200m?*
  We only address the surface signature of the AW in this sentence, and state that the surface signature of the AW (warm water) is confined to the waters where the seafloor depth exceeds 200 m. In shallower total depths, the surface water tends to be colder waters. We clarified this by rewriting as "The surface signature of the AW inflow is confined to the waters where the seafloor depth exceeds 200 m, both during fall and winter.
- *Lines 297-298. Between November and December 2019, in Figure 1e, the sea ice rose to 10%. So perhaps you should extend the time frame to the end of January 2020.*
  Indeed. We will correct and clarify this.
- *Line 330 is the 'maximum' southward 'extension' of the 'PF.'*
  The southward extension of the PF on 17 October is not the maximum southward extension. It extends further south on 6 November and arguably during the February transects.
- *Line 407. This increase in salinity during winter is not mentioned in Section 4.2.*
  Indeed. We inserted a sentence about this in section 4.2 as well.
- *Lines 410-411, in Figure 3, a northward progression of the AW/mAW is observed near the bottom.*
  Thank you for pointing this out. You are right, but it is not really a seasonal change as it occurs during October when the eddy is passing through. Between 17 October and 16 February there is hardly any change. We now comment on this in section 4.3.
- *Figures*
  - *Figure 1 The caption should state which SST and sea ice product is used, as the references to Figure 1 start in the introduction.*
    Done.
  - *Figure 2. The caption should state what the blue, orange, and yellow triangles are.*
    Done.
  - *Figure 5 could benefit from having a lower row where the standard deviation section is shown to understand in which depths the front varies more.*
    Per your main concern about section 4.2, we will revisit this figure, adding seasonal composites.
  - *DACs are integrated in the figure with altimetry? Which depth range?*
    Glider DACs cover the entire dive. For most of the mission the glider dives to within 10 meters from the seafloor.

---

## Author Response (AR1)

*Response to reviewers' and community comments on Kolås et al. EGUSPHERE-2023-2864, The Polar Front in the northwestern Barents Sea: structure, variability and mixing.*

We thank both anonymous reviewers and Dr. Maria Dolores Pérez-Hernández for the constructive comments and useful suggestions, which helped to improve the manuscript. Below we provide a point-by-point response to all comments. Reviewer's and community comments are reproduced in *italic type in red* followed by our response in regular type in black color. The future tense refers to our plan to address the comments when preparing the revised version.

**Response to Reviewer 1**

*Kolås and colleagues provide a detailed analysis of the oceanographic characteristics of the polar front in the northwestern Barents Sea (BS). The paper uses a comprehensive observational dataset (a subset of the companion paper submitted to JRL, as I understand it) obtained as part of the Nansen Legacy project with traditional and autonomous platforms (gliders), which is undeniably more detailed in terms of spatio-temporal variability, but also more complete in terms of physical variables than any previous dataset. To this end, the authors make a very valuable and serious contribution to the community. In addition, the writing and overall form are excellent and well streamlined, making life really easy for the reader.*

We thank the reviewer for the positive comments on our manuscript and for recognizing the value of our data. We are glad the reviewer found the structure of our paper satisfactory.

*This study is rather technical (which is fine!) and I find that the authors could discuss implications slightly more. The BS has long been thought a "hot spot" for marine productivity.... Due to strong vertical and/or cross-frontal mixing. However, this study confirms that the front is a place of "moderate turbulent mixing" (mainly at the surface and at the bottom) and that subsurface mixing occurs more along the isopycnals. And in fact, to the best of my knowledge, the front was never demonstrated as a significantly more productive area than the rest of the BS. The area studied is the only northern gateway to the western Barents Sea. As the front extends to the east, we might expect the same effect to occur here in the West. Interestingly, this is not the case seasonally, and from one year to another. The implications, both in terms of biogeochemical tracer and heat, are that the PF front acts more as a barrier to the AW domain (rather than a mixing machine) and to the ongoing "Atlantification". To go beyond, the AW have to subduct, transporting heat, salt and... carbon along isopycnals... which could eventually be sequestered further in the Nansen Basin.*

*Another interesting aspect of this study is that it provides the first very interesting evidence of the baroclinic structure of FP. This structure could favor baroclinic instabilities. So I can see why the authors took the time to study eddies (although they were created elsewhere). These vortices could temporarily tilt the isopycnals even further and provide additional mixing. It's fascinating that this physical feature has been studied for decades and still holds so many mysteries.*

*I found no major problems with the manuscript, which I recommend for direct publication. I recommend only very minor edits/additions for which (I think) it is not necessary to send the manuscript to the reviewers again:*

Thank you for providing useful reflections and for recommending our manuscript for direct publication. Our manuscript, particularly our results and discussion, have been revised according to the feedback provided by reviewers and community. Below we respond to your minor comments individually.

- *Please specify somewhere that altimetry provides only surface geostrophic velocities. No need extra work but you could provide existing (very few) studies that evaluated those products for quality control in the region.*
  Agreed. We now specify in section 2.6 that these are surface geostrophic velocities and have included references to Carrere et al. (2016) and Pujol et al. (2016) who evaluated these products. See snapshot below:

  > Daily and monthly sea level anomalies (SLA) and surface geostrophic velocity anomalies derived from the SLA are from
  > 205 the product SEALEVEL_GLO_PHY_L4_MY_008_047 at 0.25° resolution (Copernicus, 2023). This is a reprocessed product
  > using the DUACS multimission altimeter data processing system (Pujol et al., 2016; Carrere et al., 2016). Different altimeter

- *Figure 2: please provide brief information in the caption about transects A, D, F so that the reader so remains in the blue until Figure 9.*
  Agreed. We now mention sections A, D and F in the caption. See snapshot blow:

  > **Figure 2.** Overview of the data coverage across the PF on the sill between Hopen Trench (south) and Olga Basin (north). Isobaths are shown
  > at every 50 m between 50 m and 300 m and the 200 m isobath is highlighted in black. Red quivers are a subset of the objectively mapped,
  > divergence-free, depth-average currents presented in Kolås et al. (2023, preprint). Black quiver shows the scale. White sections (A,D and F)
  > with adjoining triangles show the location of the CTD profiles presented in Figure 9.

- *Figures 3 and 4: curious arrangement… panel labels at the bottom, colorbars squeezed in the middle. It would help to pop out the colorbars. Panels are un-scaled, you can notice through the y-axis which should be the same everywhere. This is okay and probably complex to fix, but I recommend to improve the visual. I would just use the same "frame" and leave blank where there is no data (no obligation, you probably tried already, just my suggestion).*
  Thank you for your suggestions. We agree these figures are complex. However, after careful consideration we have decided not to change them. The reason for "squeezing" the colorbars is that popping them out requires more of the width of the figure, resulting in the panels being more squeezed. The y-axis is scaled so that the height of 100m depth in one panel is equal to the height of 100m depth in all other panels. We insist that this uniform height in the vertical is a better representation when comparing different transects with varying maximum total depth. The y-axes between different panels have different heights because the individual transects differ. Some of the transects extended further south where the ocean is deeper, hence the height of that panel must be larger. We now include the following sentence in the caption: "The vertical-axis is scaled so that the height of 100 m depth is equal in all panels.".

- *Line 327, just say add "following" section, it helps to grasp the nice flow of the article. Nice transitions.*
  Agreed.

- *Figure 6: I recommend to draw some box, or arrow to describe the eddy position, structure, etc… because from isopycnals it looks like 2 eddies which are merging and it creates a bit of confusion when reading.*
  Thank you for suggesting this. We have now added a box around the eddy in panels (b), (c) and (d).

- *In the discussion I would have liked a word on the intensity of the density gradient.*
  Agreed. The gradient across the average density front presented in our study is about one tenth of the surface density front observed by Parsons et al. (1996) (0.003 kg/m³/km vs 0.05

kg/m³/km). However, the surface and subsurface fronts are two different domains and are not directly comparable. Most frontal studies in the Barents Sea studied the subsurface temperature and salinity gradients individually as they tend to be density compensating (Oziel et al., 2016; Barton et al., 2018). However, estimated from Fig. 7 in Barton et al. (2018), the 1985-2016 average density gradient across the Ludlov Saddle in the Barents Sea, at 100m depth, is approximately 0.003 kg/m³/km. We now include this in our discussion. See caption below.

> nearly horizontal, spanning almost 100 km. Nevertheless, the density gradient across the front at 100 m depth remains similar
> 450  from fall to winter, and is on average $-0.003 \pm 0.001\,\mathrm{kg\,m^{-3}\,km^{-1}}$ (Figure 3 and Table 4). This is about one-tenth of the
> density gradient across the surface density front observed by Parsons et al. (1996). However, the surface and subsurface fronts
> are two different domains and are not directly comparable. Most frontal studies in the Barents Sea have studied the subsurface
> temperature and salinity gradients individually as they tend to be density compensating (Oziel et al., 2016; Barton et al., 2018).
> However, estimated from Figure 7 in Barton et al. (2018), the 1985-2016 average density gradient across the Ludlov Saddle in
> 455  the Barents Sea, at 100m depth, is approximately $-0.003\,\mathrm{kg\,m^{-3}\,km^{-1}}$, similar to our findings.

- *I would also like a comparison of magnitude of the dissipation rate with previous estimates and/or close-by regions, besides Fer and Drinkwater (2014).*
  Dissipation rate estimates in the region are sparse. We now added comparisons to Sundfjord et al. (2007) who estimated dissipation rates across the pycnocline near Hopen Bank (see snapshot below). Other dissipation rate estimates are available along the West Spitsbergen Current, over the Yermak Plateau and north of Svalbard. However, these regions largely exhibit different forcing mechanisms for ocean mixing and a comparison to these regions falls outside the scope of this study.

> variability along the PF, depending much on the local topography. Note that we observe mixing rates in the surface layer
> 470  (upper 50 m) that are much higher than mixing rates in the lower layer throughout all of our observations, mainly exceeding
> $10^{-7}\,\mathrm{W\,kg^{-1}}$ (Figure 6d and 8). This is comparable to previous mixing rates observed on the sill between the Hopen Trench
> and the Olga Basin. Sundfjord et al. (2007) found an average dissipation rate of $4.7 \times 10^{-7}\,\mathrm{W\,kg^{-1}}$ across the pycnocline in
> May 2005, and stated that the average dissipation rates within the pycnocline were at least 5 times larger than that below the
> pycnocline.

- *If the structure of the PF is baroclinic, then the use of altimetry must be at least questioned.*
  Thank you for pointing this out. We now include a sentence about this in our discussion. See caption below.

> 485  as the SLA resolution is 0.25°. Accordingly, as the PF across the sill is baroclinic, eddies too small to be detectable by satellite
> altimetry may shed from baroclinic instabilities, feeding off the available potential energy across the front. Atadzhanova et al.

**Response to Reviewer 2**

*The authors provide a nice description of the structure of the Polar Front in the Barents Sea based on extensive ship board data (a fall and a winter cruise) as well as data from several gliders. Even though this region has been extensively studied, the study provides some novel aspects of the dynamics of the front, in particular that it is baroclinic at the sill, unlike its barotropic nature encountered further south. The paper is well-written, easy to read and does not suffer from any major issues in the methodology or interpretations. On several occasions during the read I however asked myself "So what?". Where possible it might be nice for the authors to clarify the motivation for what they are doing or what the implications of their findings could be. That being said, the baroclinic nature as well as the level of mixing at the front (weaker than at the surface and at the bottom, but quite large for the mid-water column!) are important additions to the literature that are well placed in "Ocean Science". Consequently, I can recommend this manuscript for publication after minor revisions; the minor points below do not warrant a second round at the reviewers.*

Thank you for positive feedback and for recommending our study for publication. We agree that the implications of our findings could be highlighted more, and we make an effort to do so in the revised version. This is also consistent with reviewer 1's recommendation.

*Minor points:*

*Fig. 1a This figure does not provide the important information. Most of the topographic features are not labelled and the isobaths are very hard to understand/interpret. The sill is not label or marked even though it is the key location of the paper. Fig. 11 of the authors' JGR preprint is much clearer and I'm wondering why the authors don't use a modified version of that figure here, in particular the labeling and the water depth as a color scale and not just as contour lines.*

Thank you for your suggestions. We now revised Figure 1a to better show the labelling and the water depth. See the revised version below.

[Figure]

*L143 Do you mean "southward wind" or "northerly winds"?*

Indeed, thank you for pointing this out. It has been corrected to northerly winds.

*L199 It might be nice to explicitly give the equation for C_D as a function of sea ice concentration rather than only referring to the 2005 paper.*

While it would add some convenience for some interested readers, we argue that for the reader at large the inclusion of this rather long equation (see snapshot from Lupkes and Birnbaum (2005) below) and necessary further clarifications (in the equation below, *A* is the sea ice concentration and C_dn10,i and C_dn10,w are the skin drag coefficients of ice and water, respectively. *a_r* is the aspect ratio of ice flow length vs freeboard height, which can also be expressed in terms of *A* (see equation 21 in the same paper)) could be unnecessarily distracting and complicated.

$$C_{\mathrm{dn10,e}} = 0.34 A^2 \left( \frac{(1-A)^{0.8} + 0.5(1-0.5A)^2}{a_{\mathrm{r}} + 90A} \right) + A C_{\mathrm{dn10,i}} + (1-A) C_{\mathrm{dn10,w}}. \tag{22}$$

Instead we now specify in the text which equation in Lupkes and Birnbaum (2005) we refer to, so that the interested reader can easily find it.

*L214 You give a reason, but that still leaves the question as to why you decided to do that.*

The reason for only considering the water column below 50 m depth is because the stationary front is located there. Above 50 m, we do not really capture the front in any of the ship transects, but only in a few of the glider transects. The upper 50 m and the water column below are two very different domains that are influenced by different physical processes. Here we only consider the lower layer. We now elaborate on this in the revised version of the manuscript. See caption below.

215 previous water mass definitions in literature such as Lind et al. (2018); Loeng (1991); Rudels et al. (2005). However, we have made a modification to the definition of warm Polar Water (wPW), by including only waters with potential density anomaly $\sigma_0 \geq 27.8\,\mathrm{kg\,m^{-3}}$. By excluding wPW with $\sigma_0 < 27.8\,\mathrm{kg\,m^{-3}}$ we only consider the wPW which is a mixture between AW and PW, excluding surface waters influenced to a greater extent by seasonal processes such as atmospheric heating and ice melting. The reason for this is that we miss the surface front (upper 50 m) in all ship transects, and capture it only in a few of the glider transects. The upper 50 m and the water column below are two different domains that are influenced by different 220 physical processes. Here we mainly consider the lower layer.

*L221 How are L_x and L_z estimated?*

They are estimated based on semi-variogram analysis, similar to that described in the appendix of Kolås et al. 2020. We now cross-reference this in the revised manuscript.

*L237 Is this recalculation of salinity from the sorted density profiles common oceanographic practice? Then please cite examples from the literature. Otherwise, it is a worthwhile methodological advancement that should be motivated, justified, and also advertised a bit more prominently than only with this single sentence.*

We are not aware of other examples in the literature. The reason we chose to recalculate salinity from sorted density fields in the objectively mapped sections is to reduce physical inconsistencies in the set of related variables (T, S, rho) that stem from individually mapped fields (T, S). One can objectively map the density observations, or one can calculate the density from the objectively

mapped temperature and salinity fields. We opted for the latter to make the T/S and density fields consistent. Because an objectively mapped field is not natural data, errors in T and S can propagate into density calculations and result in spurious unstable layers. A simple approach is then obtaining a physically meaningful, gravitationally stable density field and calculating salinity from this. The difference is minor. We now expand the text with our motivation, but we do not consider this a methodological advancement. See caption below.

> Because an objectively mapped field is not natural data, errors in $\Theta$ and $S_A$ can propagate into density calculations and result in spurious unstable layers. To avoid this, we calculated density and, if necessary, rearranged the values to ensure gravitationally
> 245    stable profiles. To ensure physical consistency, $S_A$ was then reproduced from these sorted density fields and $\Theta$. Note, however, that the difference between the originally mapped $S_A$ fields and the reproduced fields is minor.

*L280 Does this bias your estimation of EKE at a grid point when sea ice is present (only) at certain periods of time (which might e.g. be high, or low, EKE time periods)?*

Yes, it probably can bias our EKE estimate to some degree. EKE is likely stronger when sea ice concentration (SIC) is low versus times of dense ice cover (von Appen, 2022). Hence, removing data where sea ice concentration exceeds 15 % may cause some overestimation of the average EKE within our domain. However, the average EKE calculation within our domain is, and should be interpreted as, the average EKE within the waters with little or no sea ice, and is not representative of ice-covered waters. We now add sentences describing this: "In all EKE estimates, SLA measurements where SIC was above 15 % have been discarded. Hence the average EKE presented here is likely not representative of the region in periods when the region is mainly covered by ice. EKE is known to be stronger when SIC is low compared to times when SIC is high (von Appen, 2022).".

*L318 Why do you not calculate the gradients from each transect directly and then average the gradients to substitute the numbers currently given in L319. Note that averaged T/S will smooth (among others due to differences in the horizontal location of the strongest gradients) the gradients substantially compared to what is presumably present in each of the individual sections.*

Thank you for suggesting this. We agree and we now calculate the gradients from each transect, as suggested. The gradients from individual and composite sections are now presented in Table 4, and the mean is now calculated from the individual gradients.

*L323-330 These lines are repetitive. Consider "This reversal is discussed in the next section." And then "Simultaneously, …"*

Agreed. The transition between these two sections has been revised. See snapshot below:

> also similar during fall and winter, with the core of the current located about 10 km north of the sill. The average transport by the frontal current, calculated as the average from the individual transects, is $0.2 \pm 0.6$ Sv, where $\pm 0.6$ denotes the standard
> 350    deviation over the individual transects. The variability is relatively large, ranging between 1.1 Sv and -0.8 Sv. Negative volume transports are due to reversals of the geostrophic current (Figure 4c and 5c), which are discussed in Section 4.3. If we exclude the reversal events (3 transects), the mean eastward volume transport becomes $0.4 \pm 0.4$ Sv.
>
> **4.3   Short-term variability at the Polar Front**
>
> Between 15 and 17 October 2020 the eastward geostrophic current on the sill weakened, and reversed, flowing westward
> 355    (Figures 4 and 5). Simultaneously, the maximum southward extension of the PF increased by more than 25 km. We propose that this change was caused by an anticyclonic eddy in the Olga Basin north of the sill.

*L336-339 I can't quite follow this Eulerian vs. Lagrangian view.*

We agree that the description of the movement of the eddy can be hard to follow. Reviewer 1 suggested to draw a box around the eddy in Figure 6 to highlight the eddy. We have now implemented this in the figure and have updated the text accordingly.

*Fig. 6 caption "specified in the lower left corner" I can't see it.*

Dates are specified in the lower left corners of the subplot in panel (a). We now specify this.

*Fig. 7 caption Consider rephrasing "at B5 in fall (left) and at B7 in winter (right)".*

Agreed.

*Fig. 8 right part of figure: delta time = 1 day is a different amount of centimeters on the printed page for the upper panel (October 2020) than for the lower panel (February 2021). Consider making it equal.*

We acknowledge that there can be different views on how to best combine the spatial and temporal information in this figure. However, we would argue that since we do not compare the October and February panels directly, the differing delta time distances should not impact the readability of the figure negatively. On the other hand, if we make the bars "equal" in terms of delta time, the upper bar no longer appears next to all the October transects. That makes it difficult to see which panels it corresponds to. Consequently, we decided to keep the bar as is.

*Fig. 9 caption "in Figure 2" (a space is missing in front of the "2").*

Corrected.

*L432 "We expect the contribution"*

Corrected.

*L440 "in mid-October"*

Corrected.

*L442 "between the averaging box and the position"*

Corrected.

*L456 There are other possible explanations (L 454 "may be related to"). There might be a sea ice related bias (see comment L280). There might be an uneven distribution of events driven by external (non-climate change related) interannual variability in the 2 decades. E.g. (I'm just making up numbers/causal relations for point of illustration) EKE could be high during high NAO phases. In the first decade there were 3 years with high NAO and in the second decade there were 6 years with high NAO even though there is no long-term trend in the frequency of high NAO events.*

We agree that there could be other explanations and we now elaborate on this in the discussion (see snapshot below). A sea-ice-related bias is unlikely as more ice will potentially cause an overestimate of the EKE (see our response to L280). The sea ice cover is declining, hence there is likely more sea

ice present during the 2000-09 period compared to the 2010-19 period, and the effect of a bias likely would be to decrease the difference in EKE between the two decades.

NOA is linked to the AW transport through the Barents Sea opening (BSO), however, the mean NAO index was on average lower in the 2010-19 period than in the 2000-2009 period. In addition, local storms may affect the AW transport through BSO more than the NAO (Heukamp et al. 2023).

500   Comparing the 2000-09 decade to the 2010-19 decade suggests the EKE in general has increased in the region. The reason for this increase in EKE is not clear. The volume transport through the BSO is closely linked to the North Atlantic Oscillation (NAO) and local winds (Heukamp et al., 2023). An increased volume flux into the Barents Sea caused by faster, more energetic currents could potentially lead to an increase in observed EKE in the Barents Sea. However, there is little evidence supporting any significant difference in the volume flux through the BSO between 2000 and 2019 (Heukamp et al., 2023; Årthun et al., 2019). We suggest the increase in EKE may be related to the observed increase in oceanic heat transport into the Barents Sea,

505   and the subsequent decrease in sea ice extent, the so-called Atlantification of the Barents Sea (Skagseth et al., 2008; Årthun et al., 2012; Barton et al., 2018; Årthun et al., 2019).

*L501 "scientists" "Haakon cruise"*

Corrected.

*This study focuses on understanding the Polar Front over the sill between the Hopen Trench and Olga Basin, one of the four areas where AW meets Polar Water in the Barents Sea. The Polar Front is important for biological activity and mixing in the area. The results arise from two detailed fieldworks where hydrographic data from ship and glider sections are analyzed with altimetry, wind, and sea ice concentrations. This study is very interesting and highlights the high variability that the Polar Front has in terms of existing in location, shape, forcings, and time. The dataset used is available, and the study is relevant to the field. I suggest publishing it after some changes.*

Thank you for your valuable feedback!

*My main concern is Section 4.2, 'Polar Front structure and seasonal variability.' Here are some comments to help improve it:*

- *The way it is written reads more like a general variability than a seasonal cycle, and it finishes with an average view. Results will be better understood if the section starts with the average view and, from there, moves towards seasonality.*

  We agree. The section has now been rearranged. The composite figure (now including fall, winter and average composites, see figure in next comment) is introduced first, describing the difference between fall and winter and the average structure. Individual transects and short scale variability is moved to the following section. See caption below for the revised Section 4.2.
* * *
**315    4.2   Polar Front structure and seasonal variability**

[revised manuscript text omitted]

- *The seasonal cycle cannot be fully resolved with the available dataset. Nevertheless, a section in August could be used as representative of summer, 11 sections can be combined into a fall average section, and 4 sections can be averaged as a winter section.*
  Thank you for suggesting this. We have divided the different synoptic sections into fall and winter seasonal averages, using the specific sections you suggested. The "summer composite" sections has not been included as we only have the one August transect which is already part of the "glider transects figure".  The fall and winter composites have been included in the "average composite" figure as shown below, where (a) is fall, (b) is winter and (c) is average using all data. Note, however that (c) is average of data and not a seasonal average. It is biased towards fall as we have more data during fall than during winter. For comparison we also attach a figure of panel (c) where we simply produce an average of fall (a) and winter (b) and note that the front extend about 10 km further south in the "simple" average compared to the average composite. We comment on this in our results.

[Figure]

[Figure]

- *This section also has some minor issues, like using the term 'Atlantic-origin water' instead of 'modified Atlantic Water' as stated in Table 3.*
  The reason for using the term "Atlantic-origin water" is that it includes both wPW (as a product of AW and PW) and mAW. We agree that it was misleading and have now removed the term throughout the paper, instead referring to the correct water masses as defined in Table 3.
- *It is not said whether negative distances are located north or south of the sill in the caption of Figure 3.*
  Indeed. We have inserted a sentence clarifying this in figures 3 and 4.
- *At some point, it is stated that the velocities from altimetry match the DAC, and while that is true for October 19, the agreement is not as evident on the other two dates.*
  We agree. The barotropic geostrophic velocity calculated from SLA is a good indicator of eddies, their presence, and their strength. However, it may differ from the depth average current measured by the glider as the DAC also measures the frontal current. We now elaborate on this in the text. See caption below.

  The geostrophic velocity anomalies calculated from SLA agree well with the glider DAC (Figure 6a). Note, however, that the
  360    DAC from the glider also measures the frontal current, hence the glider DAC and the surface geostrophic velocity anomalies
  calculated from SLA are not expected to be equal.

- *Transport estimations are only given for the average section (Figure 5). You could also estimate seasonal transports or, if not, a table with the transport for each time frame.*
  We have now added a Table 4 with transport estimates from both the individual and seasonal transects, including temperature and salinity gradients across the front.
- *Overall, the text between lines 306 and 317 should be carefully revisited as they had some misleading errors.*
    - *Line 306-307. AW is separated from the surface by a warm and fresh layer; it is not cooler but fresher in the upper 60m.*
      Indeed. What we tried to communicate was that the subsurface AW core is separated from the surface layer (which is warm and fresh) by a colder interleaving layer between the AW core and the surface layer. We now clarify this in the text by writing; "The subducted AW core is separated from the surface layer by a colder, 25\,m thick, interleaving layer at about 60\,m depth.".
    - *Line 312. I don't see a cooling from December to February, as the years don't match. I see cooling on the glider dataset for November and December 2019 and the ship dataset for February 2021. These colder sections are relative to the August 2019 and October 2020 sections.*

We agree. We rewrote this, and now refer to the cooling from fall to winter, pointing to the fall and winter composite sections per your previous comment.

o *Line 314-315. In February, AW is not present (being AW defined with temperatures higher than 2ºC and salinities higher than 35.06). From what is visible in Figure 3 a, lower 2 subplots, the northern side of the Front fits better with the description of modified AW given in Table 3.*

Indeed. Only mAW and wPW is present on the Atlantic side of the front, both a result of AW cooling and mixing with PW. This is why we name them Atlantic-origin water, because it is not AW but a product from AW. We now removed the term Atlantic-origin water and specified that is was mAW and wPW.

o *Line 317. Assuming that negative distances are south of the sill, the average position lies 10km north of the sill, where the core of the positive velocities is found (Figure 5). Yet, it can reach as far as 10 km south of the sill in the 50 m depth and narrows from there to the bottom.*

Section 4.2 has now been rewritten, and the description of the average position of the front has been clarified.

- *Lines 415-416 and 420. The seasonality of the isopycnals is arguable. It says that the isopycnal tilt is flat in winter, while in Figure 3, the February sections have quite a tilt. Although the Glider sections of December have flat isopycnals, some of the October sections also present nearly flat isopycnals. So, this goes again with Section 4.2; perhaps a seasonal composite could be a better approach to assess seasonality or just blend it all under 'variability'.*

We agree that the seasonality of the isopycnals is difficult to assess based on the current figures. We now show seasonal composites to better show the change (per your previous comments), and have rewritten the discussion accordingly. See caption below.

445  PF structure. During fall, the 0°C isotherm, particularly between 50–100 m depth, is typically inclined from the horizontal and is occasionally vertical, while it is nearly horizontal during winter. Oziel et al. (2016) describe the PF position as the isotherm corresponding to the modal temperature in the region with the highest horizontal temperature gradients in the 50–100 m layer. While this may be representative of the PF location during fall, it is less suitable during winter as the temperature front becomes nearly horizontal, spanning almost 100 km. Nevertheless, the density gradient across the front at 100 m depth remains similar

450  from fall to winter, and is on average $-0.003 \pm 0.001\,\mathrm{kg\,m^{-3}\,km^{-1}}$ (Figure 3 and Table 4). This is about one-tenth of the

*Some other minor issues:*

- *Line 86 to 87. This sentence is confusing; I suggest rephrasing or avoiding mentioning Figure 1b. Here, the text refers to the data used in the study, while Figure 1b introduces a larger area.*

Agreed. We rewrote it as "An overview of the data coverage across and near the front is shown in Figure 2.".

- *Line 94. Please explain how salinity was calibrated (AUTOSAL, Portasal,other?)*

Bottle samples are analyzed at IMR with a Guildline Portasal 8410 salinometer. Salinity and conductivity values measured by the Portasal for each sample are compared with the corresponding CTD data. Following the procedure recommended by UNESCO, only data within the 95% confidence interval are used to correct the calibration of the CTD conductivity. We now add a sentence on this in the text:

"The CTD system was equipped with a SBE 32 Carousell fitted with bottles for collecting water samples at all stations. Bottle samples were analyzed using a Guildline Portasal 8410 salinometer and used to calibrate salinity."

- *Section 2.2. Two paragraphs above, it said that the cruises will be referred to as fall and winter cruises, but in this section, the names of the vessels are used. You could recall the season after the cruise name at the beginning or go with the seasonal names.*
  Agreed. We added the seasonal names.
- *Line 121. 'of the PF location (Figure 2)'.*
  Agreed.
- *Line 201. EUMETSAT OSI-SAF (2017).*
  Corrected.
- *Line 291. AW depth exceeds 200m depth? Do you mean that the entire water column is AW? or that it spreads to waters shallower than 200m?*
  We only address the surface signature of the AW in this sentence, and state that the surface signature of the AW (warm water) is confined to the waters where the seafloor depth exceeds 200 m. In shallower total depths, the surface water tends to be colder waters. We clarified this by rewriting as "The surface signature of the AW inflow is confined to the waters where the seafloor depth exceeds 200 m, both during fall and winter.
- *Lines 297-298. Between November and December 2019, in Figure 1e, the sea ice rose to 10%. So perhaps you should extend the time frame to the end of January 2020.*
  Indeed. This has been corrected. See caption below.

  and a half in December 2020 and January 2021 kept the region above the sill nearly ice free until February 2021. During the
  310    previous winter, however, the sea ice concentration in the region above the sill rose gradually from early December 2019 to mid-February 2020.

- *Line 330 is the 'maximum' southward 'extension' of the 'PF.'*
  Agreed.
- *Line 407. This increase in salinity during winter is not mentioned in Section 4.2.*
  Indeed. We inserted a sentence about this in section 4.2 as well. See caption below.

  From fall to winter, waters on both sides of the front cool, and the warm and fresh surface layer transitions into PW becoming colder and more saline (Figure 3b). During winter, PW reaches about 60 km farther south than during fall, extending to about
  330    -55 km relative to the saddle point of the sill. On the other hand, mAW and wPW below 125 m depth extend 5-10 km farther

- *Lines 410-411, in Figure 3, a northward progression of the AW/mAW is observed near the bottom.*
  Thank you for pointing this out. You are right, and we now comment on this in section 5.1. See snapshot below.

  from fall to winter (Figures 4). However, the position of the incropping of the 0°C isotherm on the north side of the sill varies by about 10 km, suggesting that the AW overflow across the sill below 100 m depth exhibits some seasonal variability. The
  440    temperature and salinity in the bottom 50 m above the sill remain above 1°C and 35 g kg$^{-1}$ between October and February,

- *Figures*
  - *Figure 1 The caption should state which SST and sea ice product is used, as the references to Figure 1 start in the introduction.*
    Done.
  - *Figure 2. The caption should state what the blue, orange, and yellow triangles are.*
    Done.
  - *Figure 5 could benefit from having a lower row where the standard deviation section is shown to understand in which depths the front varies more.*
    Per your main concern about section 4.2, we will revisit this figure, adding seasonal composites.

- *DACs are integrated in the figure with altimetry? Which depth range?*

  Glider DACs cover the entire dive. For most of the mission the glider dives to within 10 meters from the seafloor.